# NFGTRANSFORMER: EQUIVARIANT REPRESENTATION LEARNING FOR NORMAL-FORM GAMES

**Siqi Liu**[†,◇]**, Luke Marris**[†,◇]**, Georgios Piliouras**[†]**, Ian Gemp**[†]**, Nicolas Heess**[†]
[†]Google DeepMind, [◇]University College London
{liusiqi,marris,gpil,imgemp,heess}@google.com

## ABSTRACT

Normal-form games (NFGs) are the fundamental model of *strategic interaction*. We study their representation using neural networks. We describe the inherent equivariance of NFGs — any permutation of strategies describes an equivalent game — as well as the challenges this poses for representation learning. We then propose the NfgTransformer[1] architecture that leverages this equivariance, leading to state-of-the-art performance in a range of game-theoretic tasks including equilibrium-solving, deviation gain estimation and ranking, with a common approach to NFG representation. We show that the resulting model is interpretable and versatile, paving the way towards deep learning systems capable of game-theoretic reasoning when interacting with humans and with each other.

## 1 INTRODUCTION

Representing data in a learned embedding space, or representation learning, is one of the timeless ideas of deep learning. Foundational representation learning architectures (LeCun et al., 1995; Hochreiter & Schmidhuber, 1997; Vaswani et al., 2017) have provided performance and generality, bringing together researchers who used to work on expert systems targeting narrow domains. Consider Convolutional Neural Networks (CNNs): by exploiting the translation invariance inherent to images, CNNs replaced feature descriptors underpinning tasks as diverse as image classification, segmentation and in-painting, leading to paradigm shifts across the field.

One area of research that has resisted this trend is game theory. Here, we see two striking similarities to classical computer vision research. First, active research topics such as equilibrium solving (Vlatakis-Gkaragkounis et al., 2020), ranking (Balduzzi et al., 2018), social choice theory (Anshelevich et al., 2021) and population learning (Lanctot et al., 2017) all focus on specialised solutions, despite sharing the common language of normal-form games (NFGs). Second, task-specific solutions suffer from fundamental limitations. The popular Elo ranking algorithm (Elo, 2008), for instance, assigns a scalar rating to each player derived from an NFG between players. Although Elo ratings are designed to be predictive of match outcomes, they are poor predictors beyond transitive games — the Elo score is simply too restrictive a representation to reflect cyclic game dynamics. Improvements to Elo followed (Bertrand et al., 2023), but all relied on engineered feature descriptors, instead of learning. In equilibrium-solving, computing exact Nash equilibria is intractable beyond two-player zero-sum games (Daskalakis et al., 2009) yet approximate solvers are non-differentiable, take non-deterministic amount of time to converge, struggle to parallelise, and can fail. These fundamental limitations have indirect consequences too. An entire line of works in tabular multiagent reinforcement learning (RL) (Littman et al., 2001; Hu & Wellman, 2003; Greenwald et al., 2003) relied on equilibrium solving as part of their learning rules — an NFG is constructed from agents' Q-tables in each state, whose equilibrium informs subsequent policy updates. Unfortunately, reviving these ideas in the context of *deep* RL has been challenging, if not impossible, as it requires equilibrium solving as a subroutine in between every gradient update.

Indeed, we are not the first to recognise these limitations. Several recent works incorporated representation learning *implicitly*, in *narrow domains* (Marris et al., 2022; Duan et al., 2023; Vadori & Savani, 2023). We address these limitations *explicitly* and *in generality*. Our goal is to develop principled, general-purpose representation of NFGs that can be used in a wide range of game-theoretic

---

[1]The model is open-sourced at https://github.com/google-deepmind/nfg_transformer.

applications. We ask 1) which property, if any, could we leverage for efficient representation learning of NFGs without loss of generality and 2) can we expect performance in a range of game-theoretic tasks using a common approach to representing NFGs. We answer both questions affirmatively and propose NfgTransformer, a general-purpose representation learning architecture with state-of-the-art performance in tasks as diverse as equilibrium solving, deviation gain estimation and ranking.

In its most basic form, strategic interactions between players are formulated as NFGs where players simultaneously select actions and receive payoffs subject to the joint action. Strategic interactions are therefore presented as payoff tensors, with values to each player tabulated under every joint action. This tabular view of strategic interactions presents its own challenges to representation learning. Unlike modalities such as images and text whose spatial structure can be exploited for efficient representation learning (LeCun et al., 1995; Hochreiter & Schmidhuber, 1997), the position of an action in the payoff tensor is unimportant: permuting the payoff matrix of any NFG yields an equivalent game — an equivalence known as strongly isomorphic games (McKinsey, 1951; Gabarró et al., 2011). This inherent equivariance to NFGs has inspired prior works to compose order-invariant pooling functions in the neural network architecture for efficiency, albeit at the expense of the generality of the representation (Feng et al., 2021; Marris et al., 2022).

We aim to leverage this inherent equivariance of NFGs while preserving full generality of the learned representation. This implies several desiderata that we discuss in turn. First, the representation needs to be versatile, allowing for inquiries at the level of individual actions, joint-actions, per-player or for the entire game. Second, it needs to be equivariant: for any per-action inquiry, the outputs for two actions should be exchanged if their positions are exchanged in the payoff tensor. Third, the embedding function should not assume that outcomes of all joint-actions are observed — the representation should accommodate incomplete NFGs in a principled way. Fourth, the function should apply to games of different sizes. This implies that the number of network parameters should be independent from the size of the games (Hartford et al., 2016). Finally, it would be desirable if the network architecture is interpretable, allowing for inspection at different stages of the network.

In the rest of this paper, we show how our proposed encoder architecture, NfgTransformer, satisfies all these desiderata simultaneously. The key idea behind our approach is to consider the embedding function class that represents an NFG as action embeddings, reversing the generative process from actions to payoffs. Action embeddings can be suitably composed to answer questions at different granularities and allows for equivariance in a straightforward way — permutations of actions or players in the payoff tensor shall be reflected as a permutation in the action embeddings. We argue that NfgTransformer is a competitive candidate for general-purpose equivariant representation learning for NFGs, bridging the gap between deep learning and game-theoretic reasoning.

## 2 BACKGROUND

**Normal-form Games**  NFGs are the fundamental game formalism where each player $p$ simultaneously plays one of its $T$ actions $a_p \in \{a_p^1, \ldots, a_p^T\} = \mathcal{A}_p$ and receives a payoff $G_p : \mathcal{A} \to \mathbb{R}$ as a function of the joint action $a = (a_1, \ldots, a_N) \in \mathcal{A}$ of all $N$ players. Let $a = (a_p, a_{\neg p})$ with $a_{\neg p} = (\ldots, a_{p-1}, a_{p+1}, \ldots) \in \mathcal{A}_{\neg p}$ the actions of all players except $p$. Let $\sigma(a) = \sigma(a_p, a_{\neg p})$ denote the probability of players playing the joint action $a$ and $\sigma$ a probability distribution over the space of joint actions $\mathcal{A}$. A pure strategy is an action distribution that is deterministic when a mixed-strategy can be stochastic. The value to player $p$ under $\sigma$ is given as $\mathbb{E}_{a \sim \sigma}[G_p(a_p, a_{\neg p})]$. We refer to the payoff tensor tabulated according to the action and player ordering above as $G$.

**Nash Equilibrium (NE)**  Under a mixed joint strategy that factorises into an outer product of marginals $\sigma = \bigotimes_p \sigma_p$, player $p$'s unilateral deviation incentive is defined as

$$\delta_p(\sigma) = \max_{a_p' \in \mathcal{A}_p} \mathbb{E}_{a \sim \sigma} [G_p(a_p', a_{\neg p}) - G_p(a)]. \tag{1}$$

A factorisable mixed joint strategy $\sigma = \bigotimes_p \sigma_p$ is an $\epsilon$-NE if and only if $\delta(\sigma) = \max_p \delta_p(\sigma) \leq \epsilon$. We refer to this quantity as the NE GAP as it intuitively measures the distance from $\sigma$ to an NE of the game. A mixed-strategy NE is guaranteed to exist for a finite game (Nash, 1951) but exactly computing a normal-form NE beyond two-player zero-sum is PPAD-complete (Chen et al., 2009; Daskalakis et al., 2009). If $\sigma$ is deterministic with $\sigma(a) = 1$, then $\delta(\sigma)$ or equivalently $\delta(a)$ defines the maximum deviation gain of the joint pure-strategy $a$. $a$ is a pure-strategy NE when $\delta(a) = 0$.

**Permutation Equivariance**   Consider a strong isomorphism $\phi : G \to G'$ of NFGs (Gabarró et al., 2011) with $\phi = ((\tau_p, p \in [N]), \omega)$, $\tau_p : a_p^i \to a_p^{i'}$ a player action permutation and $\omega : p \to p'$ a player permutation. Elements of the transformed game $G' = \phi(G)$ are therefore given as

$$G'_{\omega(p)} \left( \tau_{\omega(1)}(a_{\omega(1)}), \ldots, \tau_{\omega(N)}(a_{\omega(N)}) \right) = G_p(a_1, ..., a_N).$$

An encoder $f : G \to (\mathbf{A}_1, \ldots, \mathbf{A}_N)$, with $\mathbf{A}_p = (\boldsymbol{a}_p^1, \ldots, \boldsymbol{a}_p^T)$ the action embeddings for player $p$, is said to be *equivariant* if

$$f(\phi(G)) = (\mathbf{A}'_1, \ldots, \mathbf{A}'_N) \text{ with } \mathbf{A}'_{\omega(p)} = (\tau_p(\boldsymbol{a}_p^1), \ldots, \tau_p(\boldsymbol{a}_p^T)) \tag{2}$$

Here we slightly abuse the notation of $\tau_p$ to operate over action embeddings. Intuitively, permutation equivariance implies that $\phi$ and $f$ commute, or $f(\phi(G)) = \phi(f(G))$. We adopt the convention that the player permutation $\omega$ is applied after player action permutations $\tau_p, \forall p$.

**Multi-Head Attention**   We describe self- and cross-attention QKV mechanisms that have become ubiquitous thanks to their generality and potential to scaling (Vaswani et al., 2017; Dosovitskiy et al., 2021; Jaegle et al., 2021). Both operations extend the basic QKV attention mechanism as follows:

$$\text{Attention}(Q, K, V) = \text{softmax} \left( \frac{QK^T}{\sqrt{d_k}} \right) V \tag{3}$$

with $Q \in \mathbb{R}^{n_q \times d_k}$, $K \in \mathbb{R}^{n_k \times d_k}$ and $V \in \mathbb{R}^{n_k \times d_v}$ the $n_q$ queries and $n_k$ key-value pairs. Conceptually, the attention mechanism outputs a weighted sum of $n_k$ values for each of $n_q$ query vectors, whose weights are determined by pairwise dot product between the key and query vectors. The output is of shape $\mathbb{R}^{n_q \times d_v}$. The inputs QKV are outputs from fully-connected networks themselves, with $Q$ a function of $x_q \in \mathbb{R}^{n_q \times d_{x_q}}$ and $K, V$ projected from the same input $x_{kv} \in \mathbb{R}^{n_k \times d_{x_{kv}}}$. The attention operation is: 1) *order-invariant* with respect to $x_{kv}$; and 2) *equivariant* with respect to $x_q$. These are the key properties we leverage in the design of the NfgTransformer to achieve its permutation equivariance property. We refer to an attention layer as self-attention when $x_q$ is the same as $x_{kv}$, and cross-attention if not. In practice, each attention layer may have $H$ attention heads performing the attention operation of Equation 3 in parallel. This enables the attention layer to aggregate multiple streams of information in one forward pass.

## 3   EQUIVARIANT GAME REPRESENTATION

While we have informally motivated the need for equivariant embedding functions, we formally state two practical implications of an equivariant embedding function that follow from a general theorem on the conditions under which two actions must have identical embeddings given an equivariant embedding function. For conciseness, we defer all formal statements and proofs to Appendix A.

**Proposition 3.1** (Repeated Actions). *If $G(a_p^i, a_{\neg p}) = G(a_p^j, a_{\neg p}), \forall a_{\neg p}$ and $f$ is deterministic and equivariant with $f(G) = (\ldots, (\ldots, \boldsymbol{a}_p^i, \ldots, \boldsymbol{a}_p^j, \ldots), \ldots)$ then it follows that $\boldsymbol{a}_p^i = \boldsymbol{a}_p^j$.*

**Proposition 3.2** (Player Symmetry). *If player $p$ and $q$ are symmetric, $f$ is deterministic and equivariant with $f(G) = (\ldots, \mathbf{A}_p, \ldots, \mathbf{A}_q, \ldots)$, then $\mathbf{A}_p$ and $\mathbf{A}_q$ are identical up to permutation.*

Proposition 3.1 guarantees by construction that repeated actions are treated identically in any downstream applications. Proposition 3.2 guarantees that player symmetry are reflected in the action embedding space which we show empirically in Section 5.3 for the NfgTransformer.

## 4   NFGTRANSFORMER

We now describe the NfgTransformer, an encoder network that factorises a payoff tensor $G$ into action embeddings via a sequence of $K$ NfgTransformer blocks (Figure 1 (Top)), each composed of a sequence of self- and cross-attention operations (Figure 1 (Bottom)). We then show concrete examples of decoders and loss functions in Figure 2 for several game-theoretic tasks at different decoding granularities, showing the generality of our approach. Finally, we discuss how the NfgTransformer naturally incorporates incomplete payoff tensors leveraging the flexibility of attention operations.

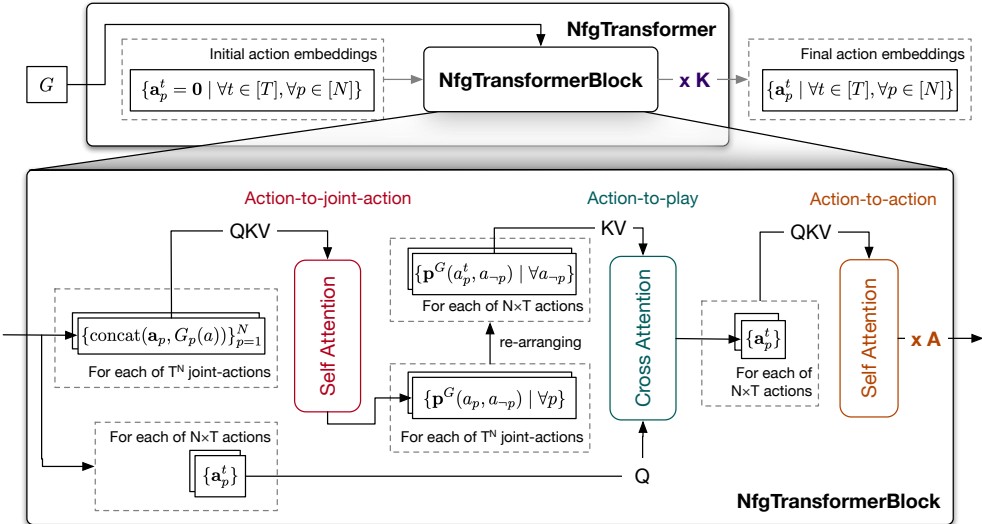

Figure 1: An overview of the NfgTransformer. The payoff tensor $G$ is encoded as action embeddings $\{\boldsymbol{a}_p^t \mid \forall t \in [T], \forall p \in [N]\}$ (Top). Action embeddings are zero-initialised and iteratively updated through a sequence of $K$ NfgTransformer blocks (Bottom). An arrow labeled with "(Q)KV" originates from a set of input (query-)key-values and terminates at a set of outputs. Each dashed box denotes an unordered set of elements of a specific type and cardinality.

## 4.1 INITIALISATION & ITERATIVE REFINEMENT

Permutation equivariance implies that action embeddings must be agnostic to how players and actions are ordered in the payoff tensor. This suggests that action embeddings across players and actions must be initialised identically. We zero-initialise all action embeddings $\mathbf{A} = \{\boldsymbol{a}_p^t = \mathbf{0} \mid \forall t \in [T], \forall p \in [N]\}$ with $\boldsymbol{a}_p^t \in \mathbb{R}^D$. Upon initialisation, action embeddings are iteratively refined via a sequence of $K$ blocks given current action embeddings and the payoff tensor $G$. Each block returns updated action embeddings via self- and cross-attention operations that we describe in turn.

**action-to-joint-action self-attention** represents a *play* of each action $a_p$ under a joint-action $a = (a_p, a_{\neg p})$ given payoff values to all players. Recall the definition of a self-attention operation, $x_q = x_{kv} = \{\text{concat}(\boldsymbol{a}_p, G_p(a))\}_{p=1}^N$, yielding one vector output per action for every joint-action. We refer to the output $\boldsymbol{p}^G(a_p, a_{\neg p})$ as a *play* of $a_p$ under the joint action $a$ and payoffs $G$;

**action-to-play cross-attention** then encodes information from all *play*s of each action $a_p^t$, with key-values $x_{kv} = \{\boldsymbol{p}^G(a_p^t, a_{\neg p}) \mid \forall a_{\neg p}\}$ and $x_q = \{\boldsymbol{a}_p^t\}$, a singleton query. This operation yields a singleton output, $\{\boldsymbol{a}_p^t\}$, as a function of all its *play*s and its input action embedding vector;

**action-to-action self-attention** then represents each action given all action embeddings. Here, $x_q = x_{kv} = \{\boldsymbol{a}_p^t \mid \forall p \in [N], \forall t \in [T]\}$. We ablate this operation (by varying $A$) in Section 5.1, showing its benefits in propagating information across action embeddings.

Within each block, equivariance is preserved given key-value order-invariance, and query equivariance properties of the attention operation. Each output embedding $\boldsymbol{a}_p^t$ is a function of its own embedding at input, its unordered set of *play*s, and the unordered set of all action embeddings.

## 4.2 TASK-SPECIFIC DECODING

The resulting action embeddings can be used for a variety of downstream tasks at different decoding granularities. We describe and empirically demonstrate three use-cases in specifics (Figure 2).

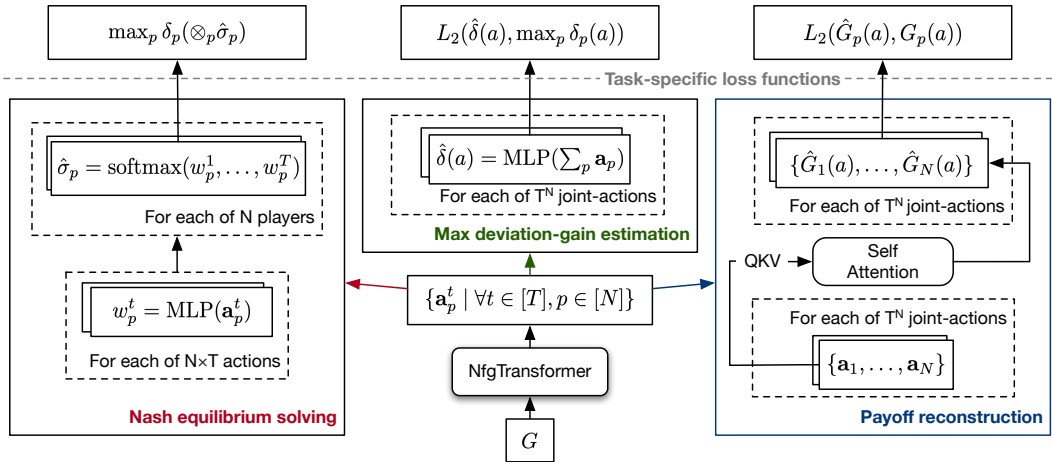

Figure 2: Example task-specific decoders and losses from general-purpose action embeddings.

**Nash equilibrium-solving** requires decoding at the level of each action, estimating a marginal action distribution for each player $\hat{\sigma}_p = \text{softmax}(w_p^1, \ldots, w_p^T)$ where $w_p^t = \text{MLP}(\boldsymbol{a}_p^t)$ is the logit for an action $a_p^t$. Here, we follow Duan et al. (2023) in minimising the loss function $\max_p \delta_p(\hat{\sigma})$ end-to-end via gradient descent with $\hat{\sigma} = \otimes_p \hat{\delta}_p$ and $\delta_p(\hat{\sigma}) = \max_{a_p' \in \mathcal{A}_p} \mathbb{E}_{a \sim \hat{\sigma}}[G_p(a_p', a_{\neg p}) - G_p(a)]$.

**Max deviation-gain estimation** decodes a scalar estimate for each joint-action $a = (a_1, \ldots, a_N), \forall a \in \mathcal{A}$. Here, we represent each joint action as $a = \sum_p \boldsymbol{a}_p$ and estimate its maximum deviation gain $\hat{\delta}(a)$ by minimising $L_2(\hat{\delta}(a), \max_p \delta_p(a)) = (\hat{\delta}(a) - \max_p \delta_p(a))^2$, with $\delta_p(a) = \max_{a_p' \in \mathcal{A}_p}[G_p(a_p', a_{\neg p}) - G_p(a)]$ the deviation gain to player $p$ under the joint action $a$.

**Payoff reconstruction** decodes a scalar for each payoff value $G_p(a), \forall p \in [N], \forall a \in \mathcal{A}$. Here we use a self-attention operation for decoding, similar to the action-to-joint-action self-attention operation in the encoder but *without* appending to action embeddings their payoff values, which are to be reconstructed. To compute a reconstruction loss, we minimise $L_2(\hat{G}_p(a), G_p(a))$, with $\hat{G}_p(a)$ a function of the action embedding $\boldsymbol{a}_p$ and the unordered set of co-player action embeddings $\boldsymbol{a}_{\neg p}$.

### 4.3 REPRESENTING INCOMPLETE GAMES

We have assumed thus far that outcomes for every joint-action of the game are observed. This is not always the case in practice as the costs of evaluating every joint-action can be prohibitive. Instead, one may infer outcomes for all joint-actions, given an incomplete NFG (Elo, 2008). With a slight modification, the NfgTransformer accommodates such use-cases in a principled way. To do so, we extend the vanilla attention implementation (Equation 3) to allow for additional binary mask vectors $\boldsymbol{m}_q \in \{0, 1\}^{n_q}$ and $\boldsymbol{m}_{kv} \in \{0, 1\}^{n_k}$ for the query and key-value inputs, indicating their validity. Equation 4 defines this masked attention operation, with $\mathbb{1}_\infty(1) = 0$ and $\mathbb{1}_\infty(0) = \infty$.

$$\text{MaskedAttention}(Q, K, V, \boldsymbol{m}_q, \boldsymbol{m}_{kv}) = \text{softmax}\left(\frac{QK^T - \mathbb{1}_\infty(\boldsymbol{m}_q \otimes \boldsymbol{m}_{kv})}{\sqrt{d_k}}\right) V \quad (4)$$

To represent incomplete NFGs, we use the Equation 4 in lieu of Equation 3 in all self- and cross-attention operations and set masking vectors accordingly. For instance, if a joint-action is unobserved, then one would set $\boldsymbol{m}_{kv}$ to reflect the validity of each *play* for the action-to-play operation. The NfgTransformer architecture is highly effective for representing actions in incomplete games and predict payoffs for unobserved joint-actions, as we show empirically in Section 5.2.

## 5 RESULTS

The goal of our empirical studies is three-fold. First, we compare the NfgTransformer to baseline architectures to demonstrate improved performance in diverse downstream tasks on synthetic and standard games. Second, we vary the model hyper-parameters and observe how they affect performance. We show in particular that some tasks require larger action embedding sizes while others benefit from more rounds of iterative refinement. Lastly, we study *how* the model learned to solve certain tasks by interpreting the sequence of learned attention masks in a controlled setting. Our results reveal that the solution found by the model reflects elements of intuitive solutions to NE-solving in games. For completeness, we discuss additional empirical results in Appendix D. For instance, we show all nontrivial equilibrium-invariant 2×2 games are embedded following structure of the known embedding space proposed in Marris et al. (2023).

### 5.1 SYNTHETIC GAMES

Table 1: We compare NfgTransformer to baseline architectures in synthetic games. Each configuration is averaged across 5 independent runs. For NfgTransformer variants (Ours), we annotate each variant with corresponding hyper-parameters ($K$, $A$ and $D$ as shown in Figure 1). We provide training curves with confidence intervals and parameter counts of each configuration in Appendix B.1.

| Model | NE (NE Gap) | | | | Max-Deviation-Gain (MSE) | | | |
|---|---|---|---|---|---|---|---|---|
| | N=2 T=16 | N=2 T=64 | N=3 T=8 | N=3 T=16 | N=2 T=16 | N=2 T=64 | N=3 T=8 | N=3 T=16 |
| Ours(D= 32,K=2,A=1) | 0.2239 | 0.1685 | 0.1344 | 0.0796 | 0.0949 | 0.5008 | 0.5206 | 0.6649 |
| Ours(D= 32,K=4,A=1) | 0.0466 | 0.1096 | 0.0892 | 0.0553 | 0.0248 | 0.3755 | 0.3679 | 0.6173 |
| Ours(D= 32,K=8,A=1) | 0.0344 | 0.0554 | 0.0484 | 0.0334 | 0.0067 | 0.2989 | 0.3582 | 0.5825 |
| Ours(D= 64,K=8,A=0) | 0.0332 | 0.0661 | 0.0636 | 0.0384 | 0.0056 | 0.1549 | 0.2848 | 0.5583 |
| Ours(D= 64,K=8,A=1) | 0.0308 | 0.0545 | 0.0478 | 0.0325 | 0.0007 | 0.0784 | 0.1830 | 0.4961 |
| Ours(D= 64,K=8,A=2) | **0.0243** | 0.0542 | 0.0437 | 0.0314 | 0.0005 | 0.0759 | 0.1942 | 0.4725 |
| Ours(D=128,K=2,A=1) | 0.2090 | 0.1665 | 0.1274 | 0.0769 | 0.0159 | 0.1922 | 0.3154 | 0.5357 |
| Ours(D=128,K=4,A=1) | 0.0429 | 0.0981 | 0.0804 | 0.0530 | 0.0013 | 0.1361 | 0.0955 | 0.4153 |
| Ours(D=128,K=8,A=1) | 0.0308 | **0.0502** | **0.0412** | **0.0297** | **0.0001** | **0.0161** | **0.0487** | **0.3641** |
| EquivariantMLP | 0.2770 | 0.2132 | **0.1431** | **0.0929** | 0.1789 | 0.8153 | 0.5433 | 0.7914 |
| MLP | 0.3905 | 0.3248 | 0.1741 | 0.1381 | 0.3854 | 0.8354 | 0.5623 | 0.7906 |
| NES | **0.0829** | **0.1635** | 0.1478 | 0.1140 | **0.0488** | **0.4860** | **0.4047** | **0.6480** |

We first evaluate variations of the NfgTransformer architecture on synthetic games of varying sizes on NE equilibrium-solving and deviation gain estimation. To generate synthetic games with broad coverage, we follow Marris et al. (2022) which samples games from the equilibrium-invariant subspace, covering all strategic interactions that can affect the equilibrium solution of an NFG. Each game's payoff tensor $G$ has zero-mean over other player strategies and Frobenius norm $\|G_p\|_F = \sqrt{T^N}$. We compare our results to baseline MLP networks with numbers of parameters *at least* that of our largest transformer variant (at 4.95M parameters), an equivariant MLP network that re-arranges actions in descending order of their average payoffs, as well as an NES (Marris et al., 2022) network that is designed for equilibrium-solving. See Appendix B.1 for details on game sampling, network architectures and parameter counts of each model. We note that the parameter count of the NfgTransformer is independent of the game size, a desideratum of Hartford et al. (2016).

#### 5.1.1 SOLVING FOR NE EQUILIBRIUM

For equilibrium solving, we optimise variants of the NfgTransformer to minimise the NE GAP $\delta(\hat{\sigma}) = \max_p \delta_p(\hat{\sigma})$ (Figure 2 (Left)). Table 1 (Left) shows our results. EquivariantMLP outperforms MLP (Duan et al., 2023), demonstrating the importance of leveraging equivariance inherent to NFGs but remains ineffective at solving this task. NES (Marris et al., 2022), equivariant by construction, significantly outperforms both MLP variants in 2-player settings but trails behind in 3-player games. The NfgTransformer is also equivariant by construction but learns to capture relevant information without handcrafted payoff feature vectors. All NfgTransformer variants, most at fewer parameter count than baselines, significantly outperform across game sizes with near-zero NE GAP.

Figure 3: Payoff prediction error averaged over all players across *unobserved* joint-actions. Results are averaged over 32 randomly sampled empirical DISC games in each game configuration.

Among the NfgTransformer variants, our results show a clear trend: increasing the number of transformer blocks (with $K \in [2, \ldots, 8]$) improves performance, especially as the game becomes large. This makes intuitive sense, as it adds to the number of times the action embeddings can be usefully refined — action embeddings at the end of one iteration become more relevant queries for the next. In contrast, the benefit of increased action embedding size is muted (with $D \in [32, \ldots, 128]$). We hypothesise that for equilibrium-solving, information related to a subset of the available actions can often be ignored through iterative refinement (e.g. dominated actions), as they do not contribute to the final equilibrium solution. Lastly, we evaluate an NfgTransformer variant that does *not* perform any action-to-action self-attention ($A = 0$). In this case, action embeddings for the *same* player do not interact within the same block and its performance is markedly worse. Of particular interests is the comparison between the variants with $A = 1$ and $A = 2$ where $A = 2$ demonstrates a benefits in the most complex games of size $16 \times 16 \times 16$ but not in smaller games. This suggests that action-to-action self-attention facilitates learning, especially in tasks that require iterative reasoning.

### 5.1.2 Estimating Maximum Deviation Gains

A related task is to determine what is the maximum incentive for any player to deviate from a joint pure-strategy $\sigma$ (or equivalently, a joint-action). This quantity is informative on the stability of a joint behaviour — in particular, if a joint pure-strategy has a maximum deviation gain $\delta(a)$ of zero, then by definition we have found a pure-strategy NE. We optimise a NfgTransformer network to regress towards the maximum deviation-gain $\delta(a)$ for every joint pure-strategy $a$, using a per joint-action decoder architecture (Figure 2 (Middle)). We report the regression loss in mean squared error of different architecture variants in Table 1 (Right). We observe that NES consistently outperforms MLP baselines, but underperforms the NfgTransformer variants as the size of the game increases.

Similar to our observations in Section 5.1.1, the number of transformer blocks played a role in transformer variants' final performance. However, it is no longer the main factor. Instead, the action-embedding size $D$ becomes critical. Variants with higher embedding size $D = 128$ can be competitive, even for *shallow* models (e.g. $K = 4$ in $16 \times 16 \times 16$ games). This can be explained by the lack of structure in the underlying game, as payoff tensors cover the full equilibrium-invariant subspace of NFGs: payoffs of one joint-action does not provide any information on the outcomes of another. To perform well, the model must learn to memorise outcomes of different joint-actions a reduced action embedding size can become a bottleneck.

### 5.2 Payoff Prediction in Empirical Disc games

What if the game class follows a *structured* generative process? This is often the case in practice and a useful representation learning method should capture any such latent structure. We turn to DISC games (Balduzzi et al., 2019) to evaluate the efficacy of NfgTransformer in this case, compared to several payoff prediction methods from the ranking literature.

**Definition 5.1** (DISC Game). Let $\boldsymbol{u}_t, \boldsymbol{v}_t \in \mathbb{R}^Z, t \in [T]$, the win-probability to action $i$ when playing against $j$ is defined as $P_{ij} = \sigma(\boldsymbol{u}_i^T \boldsymbol{v}_j - \boldsymbol{u}_j^T \boldsymbol{v}_i) = 1 - P_{ji}$ with $\sigma(x) = \frac{1}{1+e^{-x}}$ the sigmoid function.

Definition 5.1 describes a class of symmetric two-player zero-sum games where the outcomes are defined by latent vectors $\boldsymbol{u}_t, \boldsymbol{v}_t \in \mathbb{R}^Z, t \in [T]$, generalising the original definition of DISC game (Balduzzi et al., 2019) to allow for latent vectors with $Z > 1$. Payoff values under one joint-action

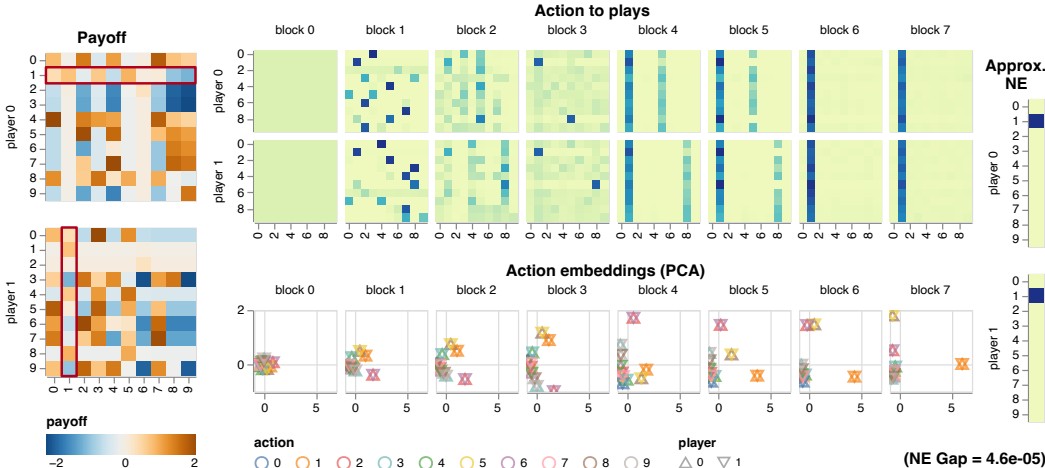

Figure 4: Visualisation of attention masks and action-embeddings at inference time on a held-out instance of Bertrand Oligopoly game whose payoff tensor is shown on the Left and the inferred NE strategy profile shown on the right (with NE GAP at near zero). The equilibrium pure-strategies for the two players are shown in red. The sequence of 8 *action-to-play* attention masks and the PCA-reduced action-embeddings at the end of each transformer block are shown in the middle.

therefore become informative for predicting the outcomes of others. The amount of information that can be inferred about one joint-action from knowing another is controlled by the latent vector dimension $Z$. The problem setting is as follows. Each algorithm is given access to an empirical game where outcomes for each joint-action is observed with probability $p$. The goal is to accurately predict the outcome to each player under *unobserved* joint-actions.

A rich body of literature have been dedicated to solving this task owing to its relevance in real-world competitions. In Go and Chess, players are classically assigned Elo ratings which are designed to be predictive of the win-probability between any two ranked players. Several improvements to Elo have been proposed since, recognising the many limitations of Elo. We compare our results to methods such as Elo (Elo, 2008), mElo (Balduzzi et al., 2018) and xElo (Bertrand et al., 2023) across different settings of the DISC game. Figure 3 shows our results in MSE averaged across *unobserved* joint-actions. NfgTransformer outperforms all baselines significantly across all settings. In particular, NfgTransformer recovered the latent variable game (i.e. $Z = 1$) near perfectly as soon as 10% of the joint-actions are observed, with an error rate an order of magnitude lower than the second best method. This result is particularly remarkable as baseline methods are designed with DISC games in mind, when NfgTransformer is not. At $Z = 8$, NfgTransformer continues to outperform, with its prediction accuracy degrading gracefully as fewer joint actions are observed. Our results suggest that NfgTransformer is highly effectively at recognising and exploiting the latent structure in games if it exists. We provide details on game sampling, masking, network architecture and baseline implementation in Appendix B.2. For NfgTransformer, outcomes of the unobserved joint-actions are masked out following the procedure described in Section 4.3. At training time, the model minimises reconstruction loss for *all* joint-actions.

## 5.3 INTERPRETING NFGTRANSFORMER

NfgTransformer explicitly reasons about action-embedding through structured attention mechanism, allowing for inspection at each stage of the iterative refinement process. We exploit this property and investigate *how* the model implements game-theoretic algorithms such as equilibrium-solving in a suite of 22 GAMUT games representing diverse strategic interactions (Nudelman et al., 2004; Porter et al., 2008) that are classically studied in the economics and game theory literature. We optimised an NfgTransformer network as in Section 5.1, but focused on $10 \times 10$ games and removed the action-to-action self-attention (i.e. $A = 0$) for a more concise visualisation. Instead of attending to multiple pieces of information in parallel, each attention layer is implemented by a single attention head

($H = 1$). We show in Appendix C that this simplified architecture is sufficient in approximating NE in games of this size to reasonable accuracy. We visualise the network on held-out games.

Figure 4 illustrates an instance of iterative NE solving in a game of Bertrand Oligopoly in action. The NfgTransformer successfully identified a pure-strategy NE (Right) with near zero NE GAP (Top-Right). Inspecting the payoff tensor (Left), we verify that player 0 playing action 2 and player 1 countering with action 1 is indeed a pure-strategy NE. Is this solution reflected in the underlying attention mechanism? Indeed, the action-to-plays attention masks (Top) appear to have converged to a recurring pattern where each action attends to the equilibrium-strategy of its co-player. This is remarkable for two reasons. First, NE by definition implies indifference across action choices — when co-players implement the equilibrium strategy, the focal player should be indifferent to its action choice. Here, this indifference played out. Second, an equilibrium solution should be stable which appears to be the case over the last few iterations of action-to-plays attentions. Zooming in on the attention masks at earlier iterations, we see that following zero-initialisation of all action embeddings, the attention mask for each action is equally spread across all its *plays*. The attention masks at the next block, by contrast, appear structured. Indeed, each action attends to the *play* that involves the best-response from the co-player. It is worth noting that this pattern of attending to one's best-response in NE-solving emerged through learning, without prior knowledge.

The action embeddings themselves also reveal interesting properties of the game, such as symmetry (Proposition 3.2). While it might not be immediately obvious that the payoff matrices in Figure 4 can be made symmetric by permuting actions, the action embeddings (Bottom) show that action-embeddings across the two players overlap exactly, thanks to the inherent equivariance of the NfgTransformer. The ordering of players and actions is unimportant — the representation of an action is entirely driven by its outcomes under the joint-actions. Here, action embeddings revealed the inherent symmetry of Bertrand Oligopoly (Bertrand, 1883).

For completeness, we offer similar analysis of the NfgTransformer applied to other game classes in Appendix C, including games without symmetry, with mixed-strategy NE as well as instances where NfgTransformer failed. Additionally, we show examples of applying NfgTransformer to out-of-distribution games such as Colonel Blotto (Roberson, 2006) that presents several strategy cycles.

## 6 RELATED WORKS

Recent works leveraged deep learning techniques to accelerate scientific discoveries in mathematics (Fawzi et al., 2022), physics (Pfau et al., 2020) and biology (Jumper et al., 2021). Our work is similarly motivated, but brings deep learning techniques to game theory and economic studies. We follow a line of works in bringing scalable solutions to game theory (Marris et al., 2022; Duan et al., 2023; Vadori & Savani, 2023), or integrating components for strategic reasoning as a part of a machine learning system (Hu & Wellman, 2003; Greenwald et al., 2003; Feng et al., 2021; Liu et al., 2022b;a). In game theory, Hartford et al. (2016) is the closest to our work and the first to apply deep representation learning techniques to cognitive modelling of human strategic reasoning using NFGs. The authors systematically outlined a number of desiderata for representation learning of two-player NFGs, including player and action equivariance as well as the independence between the number of parameters of the learned model and the size of the game. The NfgTransformer satisfies these desiderata but applies to $n$-player general-sum games. Wiedenbeck & Brinkman (2023) studies efficient data structures for payoff representation; our approach can be readily integrated into deep learning systems without any assumption on the games.

## 7 CONCLUSION

We proposed NfgTransformer as a general-purpose, equivariant architecture that represents NFGs as action embeddings. We demonstrate its versatility and effectiveness in a number of benchmark tasks from different sub-fields of game theory literature, including equilibrium solving, deviation gain estimation and ranking. We report empirical results that significantly improve upon state-of-the-art baseline methods, using a unified representation learning approach. We show that the resulting model is also interpretable and parameter-efficient. Our work paves the way for integrating game-theoretic reasoning into deep learning systems as they are deployed in the real-world.

ACKNOWLEDGMENTS

We are grateful to Bernardino Romera-Paredes for the productive discussion on the different considerations in designing an equivariant neural architecture, to Wojciech M. Czarnecki for his expertise in ranking and evaluation in games and to Skanda Koppula for his advice on optimisation techniques for the transformer architecture.

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
