## A  PERMUTATION EQUIVARIANT REPRESENTATION OF NFGS

We provide formal statements on identities in action embedding representation when using a deterministic, permutation equivariant encoder for NFGs. First, we recall the definition of a strong isomorphism between two NFGs (McKinsey, 1951; Gabarró et al., 2011).

**Definition A.1** (Strongly Isomorphic Games). Let $G$ and $G'$ be two NFGs. $G$ and $G'$ are said to be strongly isomorphic and $\phi$ a strong isomorphism if $\phi = ((\tau_p, p \in [N]), \omega)$ with $\tau_p : a_p^i \to a_p^{i'}$ a player action permutation and $\omega : p \to p'$ a player permutation such that $G'_{\omega(p)}\left(\tau_{\omega(1)}(a_{\omega(1)}), \ldots, \tau_{\omega(N)}(a_{\omega(N)})\right) = G_p(a_1, ..., a_N), \forall a \in \mathcal{A}$.

To make Definition A.1 concrete, consider the coordination and anti-coordination games shown in Figure 5. The two games are strongly isomorphic because there exists a strong isomorphism $\phi = ((\tau_1 = (a_1^1 \to a_1^2, a_1^2 \to a_1^1), \tau_2 = (a_2^1 \to a_2^1, a_2^2 \to a_2^2)), \omega = (1 \to 1, 2 \to 2))$. As aside, McKinsey (1951) calls strongly isomorphic games strategically equivalent which we discuss soon.

| $G$ | $a_2^1$ | $a_2^2$ |
|---|---|---|
| $a_1^1$ | 1, 1 | 0, 0 |
| $a_1^2$ | 0, 0 | 1, 1 |

(a) Coordination game

| $G'$ | $a_2^1$ | $a_2^2$ |
|---|---|---|
| $a_1^1$ | 0, 0 | 1, 1 |
| $a_1^2$ | 1, 1 | 0, 0 |

(b) Anti-coordination game

Figure 5: Strongly isomorphic games.

We additionally note that in the special case when $G$ is $G'$, $\phi$ is referred to as a strong *auto*morphism.

**Definition A.2** (Strongly Automorphic Game). $G$ is said to be strongly automorphic and $\phi$ a strong automorphism if $\phi = ((\tau_p, p \in [N]), \omega)$ with $\tau_p : a_p^i \to a_p^{i'}$ a player action permutation and $\omega : p \to p'$ a player permutation such that $G_{\omega(p)}\left(\tau_{\omega(1)}(a_{\omega(1)}), \ldots, \tau_{\omega(N)}(a_{\omega(N)})\right) = G_p(a_1, ..., a_N), \forall a \in \mathcal{A}$.

For instance, the coordination game (Figure 5a) is also strongly automorphic as there exists three automorphisms that recover the same game.

$$\phi = ((\tau_1 = (a_1^1 \to a_1^2, a_1^2 \to a_1^1), \tau_2 = (a_2^1 \to a_2^2, a_2^2 \to a_2^1)), \omega = (1 \to 1, 2 \to 2)) \tag{5a}$$

$$\phi = ((\tau_1 = (a_1^1 \to a_1^2, a_1^2 \to a_1^1), \tau_2 = (a_2^1 \to a_2^2, a_2^2 \to a_2^1)), \omega = (1 \to 2, 2 \to 1)) \tag{5b}$$

$$\phi = ((\tau_1 = (a_1^1 \to a_1^1, a_1^2 \to a_1^2), \tau_2 = (a_2^1 \to a_2^1, a_2^2 \to a_2^2)), \omega = (1 \to 2, 2 \to 1)) \tag{5c}$$

Note that $\phi$ is a permutation over all the players' actions which is a composition of the player and action permutations $\phi = \tau_1 \cdot ... \cdot \tau_N \cdot \omega$. Therefore $\phi$ is not a general permutation, but a structured one. We use a convention that the player permutation is applied last. Finally, we recall any permutation $\pi$ can be written uniquely as $m$ permutation orbits with $\pi = C^1, \ldots, C^m$, each operating on a disjoint (possibly singleton) subset of elements that $\pi$ operates over. Therefore $\phi$ is also a collection of permutation orbits.

Considering the coordination game again, the automorphisms (Equations 5a-5c) can be written as permutations which each consists of two orbits containing two actions each.

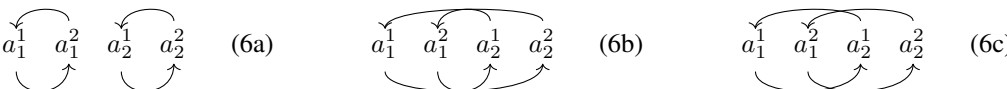

$$a_1^1 \quad a_1^2 \quad a_2^1 \quad a_2^2 \tag{6a}$$
$$a_1^1 \quad a_1^2 \quad a_2^1 \quad a_2^2 \tag{6b}$$
$$a_1^1 \quad a_1^2 \quad a_2^1 \quad a_2^2 \tag{6c}$$

**Definition A.3** (Strategically Equivalent Actions). Two actions $a_p^i$ and $a_q^j$ are strategically equivalent if there exists a strong automorphism, $\phi$ which contains $a_p^i$ and $a_q^j$ in an orbit. Equivalently, two actions $a_p^i$ and $a_q^j$ are strategically equivalent if there exists a strong automorphism, $\phi = ((..., \tau_p = (..., i \to j, ...)), \omega = (..., p \to q))$.

Again, consider the running example of the coordination game. From Equation 6a we can see that $(a_1^1, a_1^2)$, and $(a_2^1, a_2^2)$ are each strategically equivalent pairs. Furthermore, from Equation 6b we can see that $(a_1^1, a_2^2)$, and $(a_1^2, a_2^1)$ are also each strategically equivalent pairs. Therefore in the coordination games all the actions are strategically equivalent to each other.

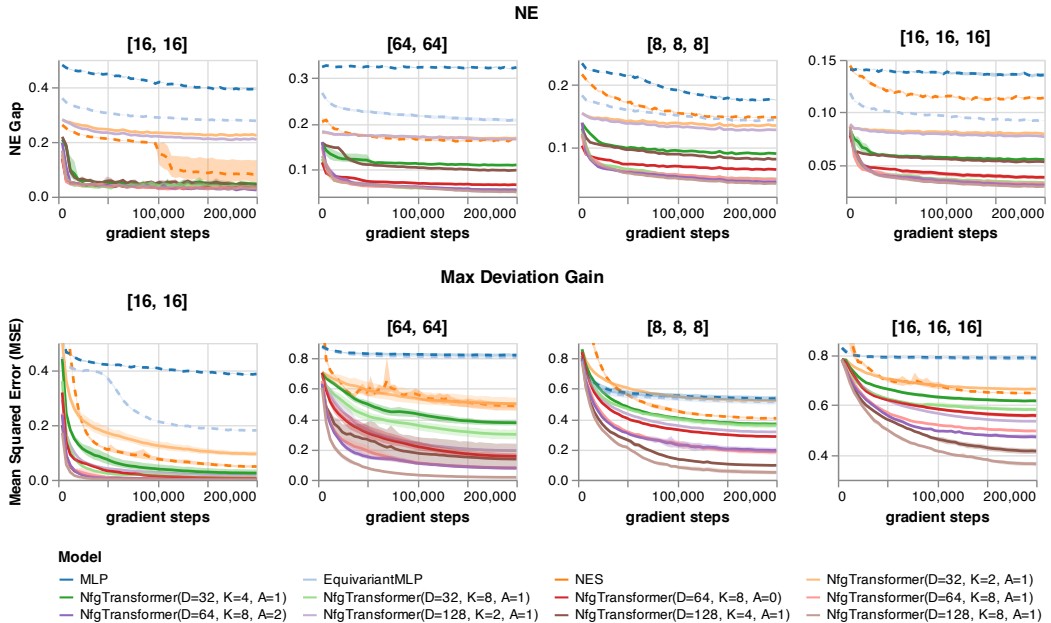

Figure 6: We compare NfgTransformer to baseline architectures in synthetic games. Results from baseline experiments are shown in dashed lines. Each configuration is averaged across 5 independent runs with shaded areas representing the confidence intervals. For NfgTransformer variants, we annotate each variant with corresponding hyper-parameters ($K$, $A$ and $D$ as shown in Figure 1).

**Theorem A.4.** *If an embedding function, $f$, is deterministic and equivariant, then strategically equivalent actions, $\boldsymbol{a}_p^i$ and $\boldsymbol{a}_q^j$, must have the same embeddings.*

*Proof.* The embedding function, $f$, is deterministic and equivariant over players and actions. Additionally, if $\phi$ is an automorphism of $G$, then $f(G) = f(\phi(G)) = \phi(f(G))$. Therefore the embeddings are also equal, $\boldsymbol{a}_p^i = \boldsymbol{a}_q^j$. $\square$

**Proposition A.5** (Repeated Actions). *If $G(a_p^i, a_{\neg p}) = G(a_p^j, a_{\neg p}), \forall a_{\neg p}$ and $f$ is deterministic and equivariant with $f(G) = (\ldots, (\ldots, \boldsymbol{a}_p^i, \ldots, \boldsymbol{a}_p^j, \ldots), \ldots)$ then it follows that $\boldsymbol{a}_p^i = \boldsymbol{a}_p^j$.*

*Proof.* If actions are repeated, there there exists an automorphism $\phi = ((\ldots, \tau_p = (\ldots, i \rightarrow j, \ldots), \ldots), \omega = identity)$. Therefore $a_p^i$ and $a_q^j$ are strategically equivalent and have the same embeddings, $\boldsymbol{a}_p^i = \boldsymbol{a}_p^j$. $\square$

**Proposition A.6** (Player Symmetry). *If player $p$ and $q$ are symmetric, $f$ is deterministic and equivariant with $f(G) = (\ldots, \mathbf{A}_p, \ldots, \mathbf{A}_q, \ldots)$, then $\mathbf{A}_p$ and $\mathbf{A}_q$ are identical up to permutation.*

*Proof.* If the game is symmetric between $p$ and $q$, there there exists an automorphism $\phi = ((\ldots, \tau_p, \ldots), \omega = (\ldots, p \rightarrow q, \ldots))$. Therefore $\tau_p(a_p^i)$ and $a_q^i$ are strategically equivalent for all $i$, and have the same embeddings, $\tau_p(\mathbf{A}_p) = \mathbf{A}_q$. $\square$

# B EXPERIMENTAL SETUP

## B.1 SUPERVISED LEARNING IN SYNTHETIC GAMES

**Games Sampling** Games are sampled from the equilibrium-invariant subspace (Marris et al., 2023; 2022), with zero-mean payoff over other player actions and a unit variance ($\sqrt{T^N}$) Frobenius tensor norm over player payoffs. To sample uniformly over such a set, first sample a game from

a unit normal distribution, $G_p \sim \mathcal{N}(0, 1)$, and then normalize.

$$G_p^{\text{equil}}(a) = \frac{\sqrt{T^N}}{Z} \left( G_p(a) - \frac{1}{T} \sum_{a_p} G_p(a_p, a_{-p}) \right) \quad Z = \left\| G_p - \frac{1}{T} \sum_{a_p} G_p(a_p, a_{-p}) \right\|_F \quad (7)$$

The benefit of this distribution is that it provides a way to uniformly sample over the space of all possible strategic interactions in a NFG of a specific shape. The equilibrium-invariant subspace has lower degree of freedom than a full NFG, freeing the neural network from having to learn offset and scale invariance. Any game can be simply mapped to the equilibrium-invariant subspace without changing its set of equilibria.

**Architecture** We provide additional technical details on the network architectures presented in Section 5.1. The baseline MLP networks are composed of 5 fully-connected layers with 1,024 hidden units each. The baseline NES architecture(Marris et al., 2022) consisted of 4 "payoff to payoff" layers with 128 channels, a "payoff to dual" layer with 256 channels and 4 "dual to dual" layers with 256 channels. Each layer uses mean and max pooling functions. All NfgTransformer model variants have $H = 8$ attention heads. Parameter counts of all model variants are reported in Table 2.

Table 2: The number of network parameters by configuration for each task. We note that the number of parameters of the NfgTransformer and the NES is independent from the size of the games. This is in contrast to fully-connected networks whose parameter counts depend on the input sizes.

| Model | # Parameter (NE) | # Parameter (Max-Deviation-Gain) |
|---|---|---|
| NfgTransformer(D= 32,K=2,A=1) | 0.15M | 0.16M |
| NfgTransformer(D= 32,K=4,A=1) | 0.31M | 0.31M |
| NfgTransformer(D= 32,K=8,A=1) | 0.61M | 0.62M |
| NfgTransformer(D= 64,K=8,A=0) | 1.10M | 1.11M |
| NfgTransformer(D=128,K=2,A=1) | 1.22M | 1.29M |
| NfgTransformer(D= 64,K=8,A=1) | 1.63M | 1.64M |
| NfgTransformer(D= 64,K=8,A=2) | 2.16M | 2.17M |
| NfgTransformer(D=128,K=4,A=1) | 2.44M | 2.51M |
| NfgTransformer(D=128,K=8,A=1) | 4.88M | 4.95M |
| EquivariantMLP | 4.76M - 16.83M | 4.99M - 20.98M |
| MLP | 4.76M - 16.83M | 4.99M - 20.98M |
| NES | 2.25M | 2.51M |

**Convergence progression** Figure 6 visualises the training progression of each model configuration, task and game size from the same experiments reported in Table 1.

### B.2 PAYOFF PREDICTION IN DISC GAMES

**Game Sampling** Following Definition 5.1, generating DISC games amounts to sampling latent vectors $\boldsymbol{u}_t, \boldsymbol{v}_t \in \mathbb{R}^Z, t \in [T]$. Any real-valued latent vectors would define a valid DISC game and we let $\boldsymbol{u}_t = \boldsymbol{n} + u$ with $\boldsymbol{n} \sim \mathcal{N}(\boldsymbol{0}, \boldsymbol{1})$ and $u \sim \mathcal{U}(-1, 1)$. We sample $\boldsymbol{v}_t$ in the same way. The shift random variable $u$ is not strictly necessary in this case, but it increases the probability that the resulting DISC game is *not* fully cyclic following Proposition 1 of Bertrand et al. (2023).

**Masking** For each sampled instance of the DISC game, with a payoff tensor of shape $[N, T, \ldots, T]$, we additionally sample a binary mask of shape $[T, \ldots, T]$ where each element follows Bernoulli($p$). Both the game payoff tensor and the sampled mask for the game tensor are provided as inputs to the NfgTransformer network. We ensure that the model does not observe the payoff values of masked joint-actions following Equation 4. During loss computation, we minimise the $L_2$ loss (Figure 2 (Right)) over all joint-actions, observed (i.e. for reconstruction) or unobserved (i.e. for prediction).

**Architecture** For all results in this section, we used NfgTransformer(K=8,A=1,D=64) with $H = 8$ attention heads for all attention operations.

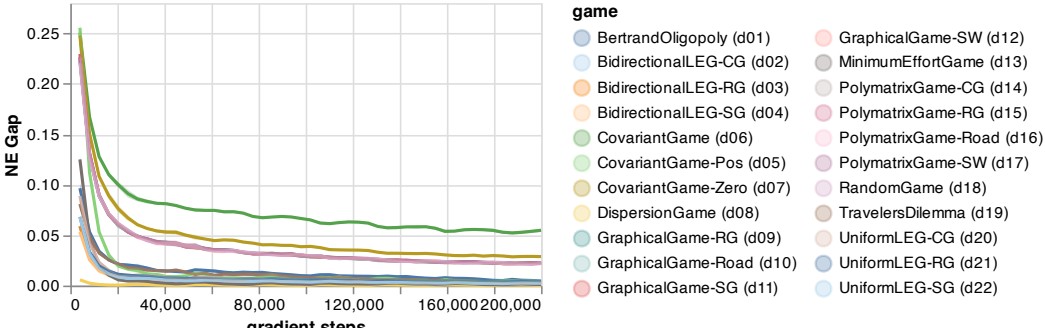

Figure 7: NE GAP reported for each of the 22 GAMUT games throughout training. We note that a single network, `NfgTransformer(K=8,A=0,D=16)`, with $H = 1$ is optimised to solve for Nash Equilibrium across all game classes.

**Baseline Solvers**  For all baseline results, we used the open-source implementation of Elo, mElo and xElo of released at `https://github.com/QB3/discrating`. For mElo and xElo, we used `n_components = 3` and the same settings as reported in Bertrand et al. (2023).

## C  INTERPRETABILITY RESULTS

We provide additional details on the empirical results in Section 5.3. Figure 7 shows that despite simplifications made in Section 5.3 for our interpretability results, the NfgTransformer remains capable of equilibrium-solving in most games to reasonable accuracy, with `CovariantGame (d06)` the most challenging game class. We show a failure case in this game class in Figure 8 (Middle) and present additional example instances where the model successfully solved for a mixed-strategy NE (Top) or generalised to the out-of-distribution game class of Blotto (Roberson, 2006). Please refer to figure caption for additional remarks on the results.

## D  THE SPACE OF 2×2 GAMES

Marris et al. (2023) introduced a subset of 2×2 normal-form games that any 2×2 game can be mapped to without changing its set of (coarse) correlated equilibria and Nash equilibria. This subset of games is called the equilibrium-invariant subset, and includes all possible nontrivial strategic interactions of 2×2 games. Properties of games such as their equilibria, permutation symmetries, and best-response dynamics can be visualized in this "map of games". We can analyse the embeddings found by the NfgTransformer by sweeping over the nontrivial 2×2 equilibrium-invariant subset.

We used the transformer architecture `NfgTransformer(K=2,A=1,D=16)` with $H = 2$ attention heads at every self- and cross-attention layer. We used an additional linear layer to reduce the action embedding dimension down to 1, per player, per action, resulting in four variables to describe the game embeddings. We trained NfgTransformer with an NE objective over the space of equilibrium-invariant subsets, and verified that the loss approaches zero. With the trained NfgTransformer, we sweep over the nontrivial 2×2 equilibrium-invariant subset, and visualize the embeddings (Figure 9).

The learned action-embeddings have a very low value (blue regions) when that action has all the mass in the NE, and very high value when the action has no mass in the NE (dark red regions). These "L" regions therefore correspond to games which have a single pure NE. The embeddings are low value (cyan regions) when the game has a mixed NE solution and occurs near the cyclic games (⊞ and ⊞). In these regions, all embeddings have to be similarly colored, as all actions are mixed. The embeddings are high value (red regions) in coordination game areas where there are two disconnected NEs (⊞ and ⊞). The borders between these regions correspond to changes in game payoffs when one action becomes become profitable than another, and as a result the NE can change drastically, and therefore so does the embedding.

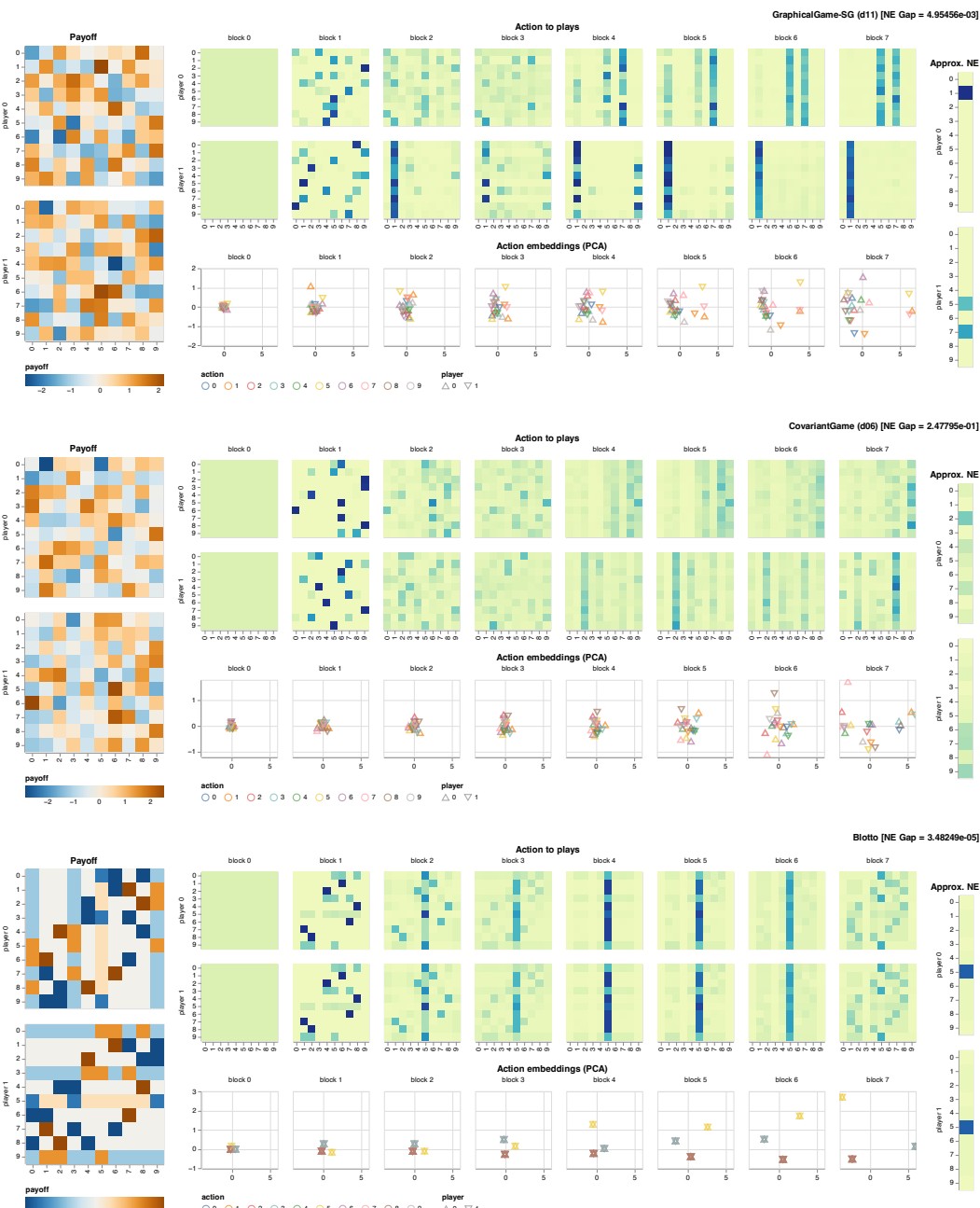

Figure 8: Here we provide additional interpretability results similar to Figure 4 but for games that are asymmetric, have mixed-strategy NE (Top) or out-of-distribution (Bottom). We additionally provide an instance where the model struggled to find an NE (Middle) where the attention masks did not appear to have converged. For Blotto (Roberson, 2006) which is a game class not seen during training, the model generalised well and identified a pure-strategy NE. The action embeddings also revealed three clusters, corresponding to the three strategically equivalent classes of actions. Note that one of the clusters corresponds to the dominant action of the two players.

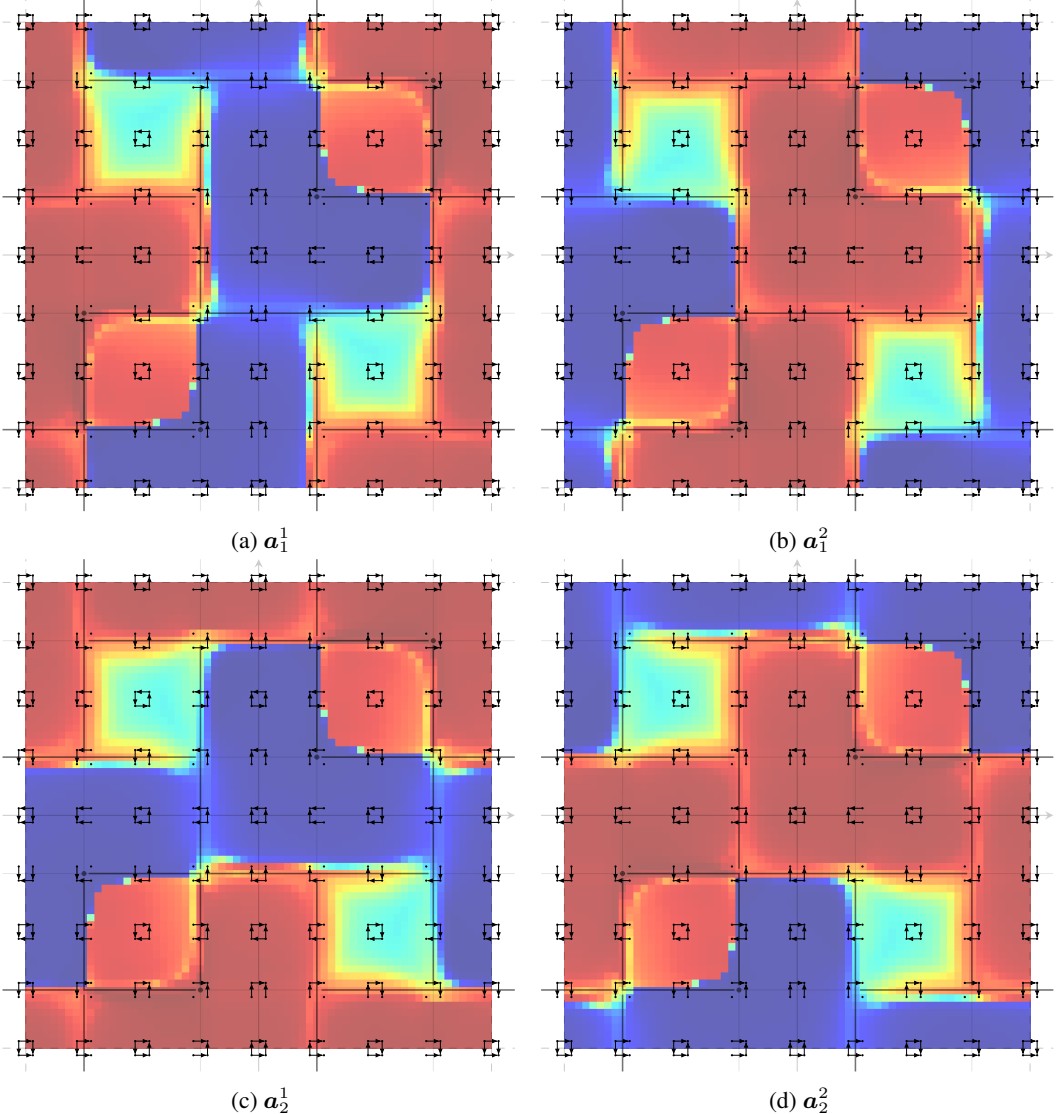

(a) $\boldsymbol{a}_1^1$

(b) $\boldsymbol{a}_1^2$

(c) $\boldsymbol{a}_2^1$

(d) $\boldsymbol{a}_2^2$

Figure 9: NfgTransformer action-embeddings over the set of nontrivial 2×2 equilibrium-invariant normal-form games, when trained with an NE objective. The embeddings found closely follow the equilibrium boundaries (dark lines). Symmetries over the space of games are respected. Symmetric games (bottom-left to top-right diagonal) have the same embeddings between players. Permutations over players (folding over the bottom-left to top-right diagonal) are consistent. Colorbar: $[-11.58$ $+11.56]$.

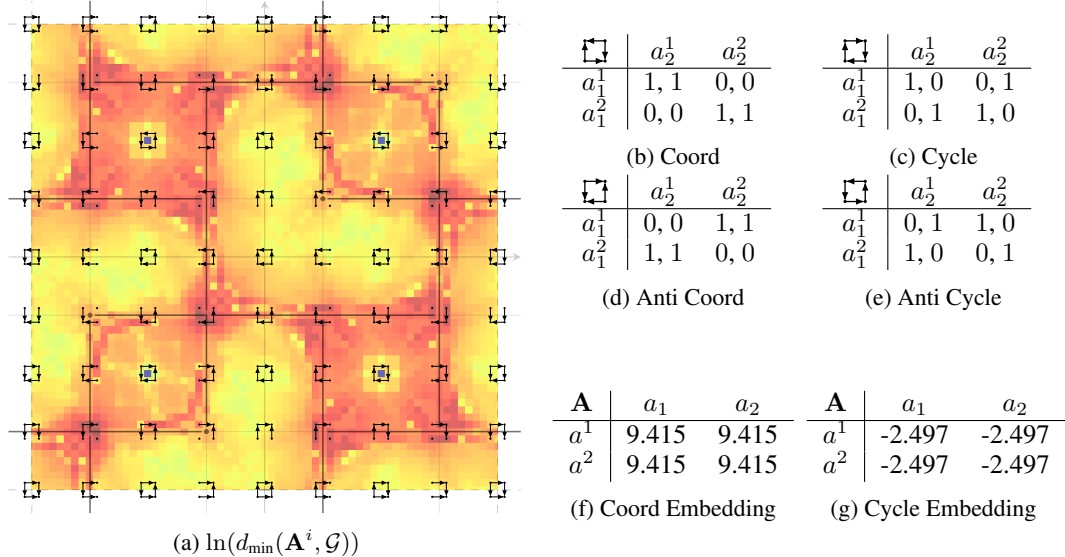

(a) $\ln(d_{\min}(\mathbf{A}^i, \mathcal{G}))$

| | $a_2^1$ | $a_2^2$ |
|---|---|---|
| $a_1^1$ | 1, 1 | 0, 0 |
| $a_1^2$ | 0, 0 | 1, 1 |

(b) Coord

| | $a_2^1$ | $a_2^2$ |
|---|---|---|
| $a_1^1$ | 1, 0 | 0, 1 |
| $a_1^2$ | 0, 1 | 1, 0 |

(c) Cycle

| | $a_2^1$ | $a_2^2$ |
|---|---|---|
| $a_1^1$ | 0, 0 | 1, 1 |
| $a_1^2$ | 1, 1 | 0, 0 |

(d) Anti Coord

| | $a_2^1$ | $a_2^2$ |
|---|---|---|
| $a_1^1$ | 0, 1 | 1, 0 |
| $a_1^2$ | 1, 0 | 0, 1 |

(e) Anti Cycle

| $\mathbf{A}$ | $a_1$ | $a_2$ |
|---|---|---|
| $a^1$ | 9.415 | 9.415 |
| $a^2$ | 9.415 | 9.415 |

(f) Coord Embedding

| $\mathbf{A}$ | $a_1$ | $a_2$ |
|---|---|---|
| $a^1$ | -2.497 | -2.497 |
| $a^2$ | -2.497 | -2.497 |

(g) Cycle Embedding

Figure 10: Subfigure 10a shows the distance to the nearest other game embedding. The embeddings produced by NfgTransformer uniquely describe the 2×2 game apart from two edge cases. Two Coordination games ( and ) have identical embeddings, and two Cycle games ( and ) have identical embeddings, each because there are strategically equivalent. Colorbar: [0.0 ▬▬▬▬▬ 4.675].

When a game is symmetric, $G_1(a_1, a_2) = G_2(a_2, a_1)$, the embeddings between players are equal. We can verify this by studying the bottom-left to top-right diagonal. When swapping the player orders, we expect the embeddings to be swapped. Swapping players is equivalent to folding over the same diagonal. Again, we can visually verify that the embeddings are swapped.

Next, we turn to the question of when the embeddings uniquely describe a game. We define a distance metric between action embeddings for game $i$ and game $j$, $d(\mathbf{A}^i, \mathbf{A}^j) = (\sum_{p \in [1,2]} \sum_{a_p \in [1,2]} (\mathbf{A}_p^i(a_p) - \mathbf{A}_p^j(a_p))^2)^{\frac{1}{2}}$, where $i, j \in \mathcal{G}$ are games sampled from a grid, which describes how close the embeddings of two games are to each other. We can also define the distance to the nearest other game within the set of considered games, $d_{\min}(\mathbf{A}^i, \mathcal{G}) = \min_{j \neq i \in \mathcal{G}} d(\mathbf{A}^i, \mathbf{A}^j)$. Using these distance metrics we can verify that $d_{\min}(\mathbf{A}^i, \mathbf{A}^j) > 0$ apart from games with strategically equivalent actions (Figure 10).

The Coordination game has identical action embeddings to the Anti-Coordination game. In this case, due to permutation equivariance, the embedding for each action, in natural language, is: "there is an action that the opponent can play which will give us both identical high payoff, and there is an action that the opponent can play which will give us both identical low payoff". Due to the equivariant property it is not possible to disambiguate between these games from the embeddings alone. By initializing the network with action labels, hinting a reconstruction method with a row of true payoffs, or permuting the payoffs by a tiny amount, would all enable disambiguation. The last strategy can be seen from the figure where slightly biased coordination games all have positive distance to their nearest other game embedding. Similarly, the Cycle game (also known as matching pennies) has identical embeddings to the Anticlockwise Cycle game. In this case, the embedding is "there is an action that the opponent can play which will me a high positive payoff and the opponent a high negative payoff, and there is an action that the opponent can play which will give me a high negative payoff and the opponent a high positive payoff". Note that the Coordination and Cycle game have distinct embeddings. These are the only 4 points in the space that can only by disambiguated up to handedness. These appear with measure zero in the equilibrium-invariant subspace.

Overall, the embeddings neatly describe and predict the known structure of 2×2 games. The theoretically predicted properties, including permutation symmetries and NE, are reproduced in this experiment.