# OpenReview forum: "NfgTransformer: Equivariant Representation Learning for Normal-form Games"
_ICLR.cc/2024/Conference — ICLR 2024 poster_

### Official Review · Reviewer_6Avg · 2023-10-16

**Soundness:** 4 excellent
**Presentation:** 3 good
**Contribution:** 4 excellent
**Rating:** 8
**Confidence:** 4

**Summary:**

In this paper, the authors aim to learn the representations of each action of every player in normal-form games. They propose a transformer-based architecture called NfgTransformer, which takes the payoffs and zero-initialized action representations as input and outputs the action embeddings. These output embeddings can be used for many downstream tasks.

In experiments, authors verify the effectiveness of NfgTransformer in Nash equilibrium solving under full-observed payoffs, and in payoff prediction when some payoffs are unobserved. Furthermore, they also conduct case studies of attention weights in small bimatrix games to demonstrate the interpretability of NfgTransformer.

**Strengths:**

1. The paper introduces a novel task: learning game representations. The acquired embeddings have the potential to enhance numerous downstream applications, making them particularly valuable in empirical game theory.
2. The attention-based architecture offers a degree of interpretability, and I found the case studies presented in Figure 5 to be particularly insightful.
3. NfgTransformer exhibits scalability as its parameters remain independent of the number of actions

**Weaknesses:**

The presentation can be further improved:

(a) The description of the three attention mechanisms in Section 4.2 is too brief. I believe it would be beneficial to introduce more mathematical expressions.

(b) Also, in Section 4.2, I find the name of the first attention mechanism to be misleading. I would suggest using "joint-action-to-player self-attention" rather than the original "player-to-coplayer self-attention."

(c) I'm having difficulty discerning the results presented in Figure 3. Perhaps using a table would be clearer than the loss curves.

**Questions:**

1. At the end of Section 4.4, you mentioned, "During training, invalid joint-actions are simply masked out from the set of values..." How are these invalid joint-actions masked out? Is this achieved by reducing the corresponding pre-softmax scores significantly?
2. In Section 5.2, how are the instances of DISC games generated? How do you train your model?

---

> ### Author Response · Authors · 2023-11-16
>
> We thank the reviewer for recognising the significance and innovation presented in our work. We hope our significant revision to the main text and figures following several of your suggestions will further improve clarity.
>
> **Q1: clarity of presentation in the method section as well as Figure 3**
> We fully agree with your assessment and we took on board several of your suggestions in our revision (see updated PDF, with major changes highlighted in blue). We have fully reworked Figure 1 and Figure 2 as well.
>
> Specifically,
> * We introduced precise mathematical notations where possible in describing our method (in Figures and in text).
> * We renamed “player-to-coplayer self-attention” to “action-to-joint-action self-attention”. This is slightly different from your suggestion but it reflects the fact that in the self-attention module, queries are the action of each player, and the key-values attended over are the joint-action (and their respective payoffs).
>
> Regarding Figure 3, we tried to provide thorough ablation results here as they differ based on the nature of the tasks, the game sizes and the model hyperparameters. We are actively trying out other ways of presenting these results and perhaps a table would indeed work best. We could still provide the learning curves (showing the rate of convergence) in an appendix section for interested readers. We will update with revision when ready.
>
> **Q2: How are these invalid joint-actions masked out?**
> We unfortunately glossed over this detail in our initial submission. We have now included the precise implementation of this operation in Equation 4.
>
> Indeed, the implementation amounts to enforcing invalid combinations of query-key-values to have negative infinity in their pre-softmax logits, effectively removing their contribution in the outputs.
>
> **Q3: how are the DISC games sampled? And how is the model trained?**
>
> We provided technical details in how the DISC games are generated in Appendix B.2. but we failed to signpost it in the main text clearly. We do so in our revision now.
>
> In short, we sample latent vectors (with different dimension Z) underlying a DISC game from a standard normal distribution and add to them a noise vector sampled from a uniform distribution.
>
> For each sampled instance of the DISC game, we additionally sample a mask (such that each joint-action is kept with probability $p$) and we pass both the sampled game payoff tensor and the sampled mask to the NfgTransformer model, which returns a set of action embeddings, without observing the outcomes of invalid joint-actions (we ensure this via Equation 4). We then decode from action embeddings predicted payoff values for every joint-action and simply minimize the reconstruction loss given the ground truth full-payoff (as described in Figure 2 (Right)). The loss is minimised end-to-end via gradient descent.
>
> We have added a description of the above training procedure in Appendix B to clarify our settings for future readers.

---

> > ### Author Response · Authors · 2023-11-17
> >
> > A quick update: we have now replaced Figure 3 in the main text with a Table following your suggestion. We think it is indeed much more readable. Thank you! Figure 3 is now moved to the Appendix and readers could still refer to it for completeness.

---

> > ### Comment · Reviewer_6Avg · 2023-11-20
> >
> > Thank you for your response and the significant revision. The presentation is now clearer than before, and the clarifications have addressed my questions. I will maintain my score.
> >
> > Regarding the paper's presentation, I have some further personal suggestions:
> > 1. It seems that the Primary Area could be 'unsupervised, self-supervised, semi-supervised, and supervised representation learning' or 'algorithmic game theory' to better characterize your paper.
> > 2. The key contribution of your paper should be learning the game representation in an **explicit** way. This is because, in previous works on learning equilibrium solvers using neural networks (Feng et al., 2021; Marris et al., 2022; Duan et al., 2023a;b), they already obtain game representations **implicitly**. The intermediate result of their architecture before the last layer can be seen as game representations.
> > 3. Considering that previous works have already learned game representations implicitly, you can describe the advantages of learning game representations in a more convincing way. Afterward, you can delve into a more detailed explanation of why we need to learn the representations explicitly.

---

> ### Author Response · Authors · 2023-11-20
>
> Thank you for your quick response and for reconfirming your positive assessment of our work. A few quick responses below:
>
> **Q1. choice of primary area**
>
> We have indeed hesitated between the two choices. As we study the representation learning for a *novel* data modality previously under discussed, we tried to be non-committal in our choice. The "representation learning" category indeed seems more appropriate. "Algorithmic game theory" perhaps hints at solving specific game theory tasks which is a secondary effect of our paper not a primary goal.
>
> We can't seem to make the change now to "representation learning", but we will follow up with ICLR to make the switch if possible if/when the paper gets accepted.
>
> **Q2/Q3: explicit vs. implicit representation learning**
>
> This is an excellent remark and we will rephrase this comparison in our introductory paragraphs to make this comparison stand out more.
>
> *A few quick remarks on this point:*
>
> It is true that prior works *implicitly* relied on representation learning, however, they are either lossy (e.g. Feng et al., 2021; Marris et al., 2022) due to the use of order-invariant pooling functions (see Marris et al., 2022 Appendix C.1 for a list of candidate pooling functions) or inefficient (e.g. Duan et al., 2023a;b), using MLPs which do not exploit the permutation equivariance properties of NFGs.
>
> In fact, for the specialist pooling-based architectures, one could simply handcraft pooling functions *for some tasks* such that no representation *learning* is needed at all. For `max-deviation-gain` prediction, one could imagine designing pooling function that directly extract max-deviation-gain from the payoff tensor and expose them in the penultimate layer of the network (indeed, without such customisation, NES seemed to struggle in the `max-deviation-gain` prediction task more than it struggled in the NE-solving task). This would not in our view qualify as representation learning. However, the boundary is blurred across tasks: for equilibrium-solving, design of such pooling functions wouldn't be possible and representation learning would be implicit indeed as noted in your comment.

---

> > ### Author Response · Authors · 2023-11-20
> >
> > We have now updated our main text revision and made the contrast in our view to representation learning to prior works more explicitly (paragraph 3 and 4) following your suggestions. Again we would like to thank the reviewer for helping us improving our work significantly throughout this process.

---

> > > ### Author Response · Authors · 2023-11-20
> > >
> > > As we are heading towards the end of the discussion period, we are wondering if you would reconsider the initial assessment of the "presentation" scoring (currently at 2) given our latest revision? Thank you for your continued support for our work!

---

### Official Review · Reviewer_s4as · 2023-11-06

**Soundness:** 2 fair
**Presentation:** 3 good
**Contribution:** 2 fair
**Rating:** 5
**Confidence:** 4

**Summary:**

This paper considers the permutation equivariance in normal-form games (NFGs) in terms of player action and players. The authors claim that there is inherent equivariance in NFGs – “any permutation of strategies describes an equivalent game.” Then they design an NfgTransformer to leverage this equivariance. They conduct comprehensive experiments show their method significantly outperformes strong baselines in a range of game-theoretic applications.

**Strengths:**

This paper proposes to discover the "inherent symmetry" in game theoretical problems using equivariant neural networks. This idea is impressive, timely, and promising.

The authors design an NfgTransformer to meet their aim. I find using transformer reasonable.

They also conducted experiments covering a wide range of cases.

**Weaknesses:**

My concerns are as follows:

- I am worried about the claim that permutation equivariance is "inherent" in normal-form games either in terms of action or player. This is not obvious to me. Please prove this. Also, such permutation equivariance is a quite strong assumption, and leads to a significantly restrictive application domain.

- It is not clear to what extent the proposed NfgTransformer is different from a vanilla one which has naturally been permutation equivariant. The authors did not clearly present the architecture of NfgTransformer - I could not validate the novelty in terms of model design.

- The authors did not give any theoretical guarantee - whether the proposed method secure the desired permutation equivariance, whether the method can reach the desired Nash equilibrium, how fast would the algorithm converge, whether the generalisability of the proposed method (as a learning algorithm) is satisfiable, whether the proposed method has any theoretical advances, etc. Before answering this question, I was quite hesitant about the proposed algorithm.

- I appreciate the empirical results. However, game theory has a very wide range of applications - I do not think experiments in a few applications could justify an algorithm which is claimed to have advances in general cases.

**Questions:**

Please address the above.

---

> ### Author Response · Authors · 2023-11-16
>
> We thank the reviewers for pointing out potential sources of confusion in our presentation. We hope our revision to the main text would help clarify our arguments.
>
> **Q1: Is permutation equivariance inherent to normal-form games?**
>
> By “inherent equivariance of NFGs” we mean that action or player permutations applied to the payoff tensor of an NFG do not fundamentally “change” the game. Note that this general property of NFGs has been observed and formally characterised in [1-2].
>
> This is *not* an assumption that we make, or a property discovered in our work but a general observation that holds for all NFGs and this is why we can leverage this property in our design without loss of generality.
>
> As a simple example, consider $G$, a payoff matrix that describes a prisoner’s dilemma. I could permute the two actions for the row-player and obtain $G’$, which is “strategically equivalent” or strongly isomorphic to $G$.
>
> | $G$   	| $a^1_2$  	| $a^2_2$  	|
> |:--:|:--:|:--:|
> | $a^1_1$ 	| (+1, +1) 	| (-1, +2) 	|
> | $a^2_1$ 	| (+2, -1) 	| (-1, -1) 	|
>
> | $G’$   	| $a’^1_2$  	| $a’^2_2$  	|
> |:--:|:--:|:--:|
> | $a’^1_1$ 	| (+2, -1) 	| (-1, -1) 	|
> | $a’^2_1$ 	| (+1, +1) 	| (-1, +2) 	|
>
> If I were to encode G1 and G2 using NfgTransformer, I would expect the following identity in the action embeddings: $\mathbf{a’}^2_1 = \mathbf{a}^1_1$. If I were to use a MLP network then I would not have this guarantee.
>
> **Q2: how does NfgTransformer differ from vanilla transformer?**
>
> We have significantly reworked our method description as well as Figure 1 and Figure 2 that describe the model architecture in specifics. The only commonality between our proposed model and vanilla transformer is that both architectures use self-/cross-attention operations as fundamental building blocks. The NfgTransformer is otherwise distinct from vanilla Transformer in how these components are applied and used.
>
> We hope our revision to the main text and figures could help verify the novelty of our model design.
>
> **Q3: Is permutation equivariance guaranteed?**
>
> The permutation equivariance property is guaranteed by construction. In our revision, we have slightly expanded on the background section on the attention mechanism which is equivariant with respect to queries (and order-invariant with respect to key-values). In the revised Figure 1 (Bottom), we make clear what are the queries and key-values to each attention operation, and show how permutation equivariance is guaranteed by careful design.
>
> Empirically, our interpretability results (Figure 5, action embedding visualisation) also confirm that Proposition 3.2 holds, which is a property for any permutation equivariant encoding function (which we proved in Appendix A).
>
> **Q4: Are there any theoretical guarantees to the proposed architecture?**
>
> We do not have any bounds on the rate of convergence of our model as it’s based on deep learning neural networks architecture and such guarantees are difficult to prove in general. However, we do show significantly better empirical results than several comparable baseline methods, and our Figure 3 thoroughly ablates different model hyper-parameters, showing which parameters affect the rate of convergence in different tasks at each game size.
>
> In terms of theoretical guarantees, we did prove (in Appendix A) the precise conditions under which one could expect identities in the action embeddings, for any deterministic and equivariant embedding function. NfgTransformer is deterministic and equivariant and these identities must hold. Two straightforward corollaries of this theorem are stated in Section 3.
>
> **Q5: are the empirical results sufficient in making the generality claim?**
>
> We think the best way to demonstrate the generality of any representation learning technique is to demonstrate that the same representation learning approach can enable strong results in a number of downstream tasks. In domains such as computer vision one would be expected to provide results on reconstruction, classification among other benchmark tasks.
>
> Here, we showed strong empirical results in key tasks such as equilibrium-solving, deviation gain estimation and payoff prediction. We chose these tasks because they operate at different decoding granularities (see our updated Figure 2), and cover many of the interesting use-cases of normal-form game theory. We believe these support our claim on the generality of our approach to representing NFGs.
>
>
> [1] J. C. C. McKinsey.11. ISOMORPHISM OF GAMES, AND STRATEGIC EQUIVALENCE, pp. 117–130.   Princeton University Press, Princeton,  1951.
>
> [2] Joaquim Gabarŕo, Alina Garcıa, and Maria Serna. The complexity of game isomorphism. Theoreti-
> cal Computer Science, 412(48):6675–6695, 2011.

---

> > ### Author Response · Authors · 2023-11-21
> > **Discussion period ending soon**
> >
> > Dear reviewer s4as,
> >
> > As we are approaching the end of the discussion period we are wondering if you have had a chance to review our significant revision and our responses above? If so, please do let us know if our revision has addressed your questions and concerns.
> >
> > Thank you!

---

> > > ### Comment · Reviewer_s4as · 2023-11-22
> > >
> > > Thanks for your response.
> > >
> > > My first concern is cleared (about whether normal-form games are permutation equivariant).
> > >
> > > I still see weaknesses in the novelty of architecture design, theoretical analysis, and empirical verification.
> > >
> > > I now recognise the contribution in proposing to learn game representation, but the development/verification of the idea is not in the perfect form.
> > >
> > > I will increase my score to 5.

---

> > > > ### Author Response · Authors · 2023-11-22
> > > >
> > > > We are glad to see our revision has addressed several of your initial concerns (e.g. clarity and motivation) and we appreciate your taking the time to review our revision. Thank you!
> > > >
> > > > > novelty of architecture design, theoretical analysis, and empirical verification.
> > > >
> > > > A few quick comments regarding these outstanding points of concern:
> > > > 1. our work is the very first study on principled representation learning techniques for NFGs and our architecture design is novel in its own right;
> > > > 2. our model comes with permutation equivariance guarantees. We are also the first to provide a theoretical analysis on the precise conditions under which two actions of an NFG must be embedded identically (see Appendix A for a full proof, and see Figure 4 for the symmetry identity proposition playing out with our architecture);
> > > > 3. we provided strong empirical results, outperforming prior works on equilibrium solving, ranking, deviation gain estimation by a very significant margin using a *unified representation learning technique*. Baseline methods we compared to are expert systems optimised for each task (NES for equilibrium-solving, Elo/xElo/mElo for ranking).
> > > >
> > > > We plan on open-sourcing our model implementation upon publication too and these results would certainly help enable future research in this direction. We remain available to address any outstanding concerns until the end of the discussion period.
> > > >
> > > > We believe ICLR would be an excellent venue for this work. Please let us know if there is any room for further improving your final assessment of our work. Thank you for your understanding!

---

### Official Review · Reviewer_pter · 2023-11-06

**Soundness:** 2 fair
**Presentation:** 1 poor
**Contribution:** 1 poor
**Rating:** 8
**Confidence:** 3

**Summary:**

The submission proposes a transformer architecture for learning normal-form games. They evaluate this architecture on a variety of game-related tasks, including solving for Nash equilibria and predicting deviation payoffs.

**Strengths:**

Regarding originality, I don't know of any paper that has done what the submission seems to be doing.

**Weaknesses:**

### First sentence

I'm going to start off by discussing the first sentence of the introduction:

> A number of celebrated results in artificial intelligence (AI) have appealed to game-theoretic principles during training, evaluation and ranking (Silver et al., 2016; Lanctot et al., 2017; Vinyals et al., 2019; Omidshafiei et al., 2019; Liu et al., 2022b; Perolat et al., 2022).

The way this sentence is written leaves the reader with the impression that the citations are examples of celebrated results that have appealed to game-theoretic principles. Let's go through them and examine whether this claim is true. I agree that Silver et al. (2016) is a celebrated result. However, it is not clear to me that it uses any game-theoretic principles during training (it uses supervised learning + self-play) or evaluation/ranking (it uses elo rankings). Lanctot et al. (2017) is not a result at all -- it is a paper that advocates for combining double oracle and deep RL. Vinyals et al. (2019), Omidshafiei et al. (2019), Liu et al. (2022b), Perolat et al. (2022) all use some level of game-theoretic justification, but none live up to the very high standard implied by the term "celebrated". One also cannot help but notice that all six of the cited papers are DeepMind papers. While it is certainly the case that DeepMind has made important contributions to RL in games (and perhaps more so than any other group), citing only DeepMind papers here is completely unjustifiable. Arguably, the two most significant sets of successes that directly appeal to game-theoretic principles are those concerning poker (DeepStack/Libratus/Pluribus/ReBeL) and No-Press Diplomacy (Diplodocus). That the submission choses to cite DeepMind papers as opposed to more-relevant non-DeepMind papers to support their claim gives the impression that it is attempting to push a narrative rather than give an honest perspective (or, alternatively, that the submission is simply uninformed). One might argue here that I am being too pedantic about a somewhat boilerplate sentence in the introduction that is not closely related to the submission's contributions. But I think it is important that the submission not mislead the reader.

---

### Motivation of paper

After reading the introduction, I am not quite sure I fully grasp the problem that the submission is trying to solve. The submission abstractly states "We consider the problem of bringing game-theoretic reasoning to deep learning systems and, conversely, using deep learning techniques to solve challenges in game theory" but does not immediately clarify why would would want bring game-theoretic reasoning to deep learning systems or what challenges the submission is going to solve in game theory. It further states "In its most basic form, strategic interactions between players are formulated as NFGs where players simultaneously select actions and receive payoffs subject to the joint action. Strategic interactions are therefore presented as payoff tensors, where values to each player are tabulated under every joint action. This tabular view of strategic interactions presents its own challenges to representation learning." Under the presumption that doing representation learning on a normal form game is important, I agree there are some representation learning-related challenges, but the submission hasn't really explained why we would want to be doing this in the first place. The submission then lists some game-theory related questions "Examples of such enquiries permeate different communities within game theory: given sets of actions, what would be
an equilibrium strategy (Greenwald et al., 2003; Marris et al., 2022; Duan et al., 2023a;b)? How does the efficiency of the system degrade due to individual selfishness (Koutsoupias & Papadimitriou,1999)? How might we cluster actions considering transitivity and strategy cycles (Czarnecki et al., 2020)? Given outcomes for some joint actions, can one predict payoffs for the others (Balduzzi et al., 2018; Bertrand et al., 2023; Vadori & Savani, 2023)? To what extent can we reduce the dimensionality of a class of NFGs (Marris et al., 2023)?" This gives the reader the impression that these are some of the questions the submission is interested in. But the big question of *Why we need to learn a representation of the game to do this* remains unaddressed -- it is unclear to the why problems such as "what would be an equilibrium strategy" ought not to be addressed from a normal-form representation. The submission proceeds to list desiderata for a unified representation for solving these tasks.

---

### Description of methodology

The submission refers the reader to Figure 1 to understand the architecture that they're proposing, but I find the figure not very easy to grok. What does it mean when two arrows point at the same block?

The submission describes the architecture in Section 4, but I could not figure out what the architecture was being trained to do. I also still do not understand what the downstream use case of this architecture is.

---

### Experiments

In Section 5.1.1, the submission says this "For equilibrium solving, we optimise variants of the NfgTransformer to minimise the NE GAP
() = maxp p() directly." How is the submission performing this optimization? How are we recovering a NE strategy from the action embeddings? These questions are central to the paper, but seem to be unanswered in the text. Furthermore, why should we want to solve NFGs with this architecture instead of using a classical method? Without discussing this question, the reader is left wondering why they should care about these experiments.

In Section 5.1.2, the submission says this "We optimise a NfgTransformer network to regress towards the maximum deviation-gain () for every joint pure-strategy (deterministic) , using a per joint-action decoder architecture (Figure 2)." Again, it is totally not obvious how the submission is doing this based on the text in the paper. Furthermore, the submission again has no discussion about why we should want to use an architecture instead of a classical method for this task (which in this case just amounts to matrix multiplications).

Section 5.2 studies a kind of payoff table completion task. Here it seems reasonable that one might actually want to use an architecture of the kind the submission describes. However, the submission still neglects to disclose how the the architecture was trained.

**Questions:**

What's a regular isomorphism as opposed to strong isomorphism in the context of games?

---

Overall, I think the submission needs to be re-written to make it more clear what problem it is trying to solve and how it is training its architecture.

---

> ### Author Response · Authors · 2023-11-16
>
> **Q1: choice of citations in the introduction and motivations behind the work.**
>
> You are right (and not being pedantic) to point out the issues with the initial motivations and citations used. We regret the boilerplate sentences used and have reworked the entire introduction to have a clearer motivation and have updated citations to not omit important work in the field.
>
> We hope our revision and more relevant citations would help clarify the motivation behind our work.
>
> **Q2: Lack of clarity of the method description.**
>
> We took on board these constructive criticisms and have revised Figure 1 and Figure 2 entirely, with precise notations throughout. We have also updated our method description in text accordingly, preferring precise notations where possible. In particular, the three downstream applications that we studied are now clearly described, with specific loss functions shown in Figure 2 and in text.
>
> We hope our revision brings clarity to our method descriptions.
>
> **Q3: The “how” and “why” behind the experiments.**
>
> We hope our earlier responses and our revision to the main paper provides sufficient clarity now. All three empirical experiments are now described more precisely in Figure 2. We train these models end-to-end via gradient descent, minimising the loss functions described in Figure 2 (Top) and in text. Equation 4 now describes precisely how masking is implemented in our experiment.
>
> **Q4: regular isomorphism vs strong isomorphism for NFGs.**
>
> [1] offers more precision definition of isomorphic games and strongly isomorphic NFGs. In short, isomorphic games would preserve the “preference ranking” of actions when players and actions are permuted. Strongly isomorphic games preserve the exact payoff value before and after the permutation.
>
> For example, rock-paper-scissors and paper-scissors-rock are strongly isomorphic because I can permute the former to recover the exact payoff matrix of the latter (i.e. payoff values match, not just in the preference ranking of the actions).
>
> We provide additional details in Appendix A, Figure 6 on the notion of strong isomorphism.
>
>
> [1] Joaquim Gabarŕo, Alina Garcıa, and Maria Serna. The complexity of game isomorphism. Theoreti-
> cal Computer Science, 412(48):6675–6695, 2011.

---

> > ### Comment · Reviewer_pter · 2023-11-20
> > **Message to Authors**
> >
> > Thanks so much for your response and revision! I'm sorry my response has been so delayed. It is regrettable that the discussion period is so short. I will be sure to respond in detail in the next day or two.

---

> > > ### Author Response · Authors · 2023-11-20
> > >
> > > Thank you for the update!
> > >
> > > We look forward to hearing from you soon and please feel free to point out any further sources of confusion as they arise and we will be sure to respond in a timely fashion.

---

> > > > ### Comment · Reviewer_pter · 2023-11-21
> > > > **Reply to Authors**
> > > >
> > > > I read through the revised version of the text. I appreciate the changes to the introduction and the more detailed description of the training process. One thing I feel like I'm still not quite understanding is the motivation of the problem. In a previous version of their response, the authors stated:
> > > >
> > > > > In short, NfgTransformer is preferable to classical solvers as they can be used as another neural network component within a deep learning system that’s fully differentiable, can be parallelised and takes deterministic time. This property can unlock lines of research that could benefit from baking in equilibrium concepts in the agent learning-rules [1-3]. For ranking, as suggested by the reviewer, NfgTransformer provides a much richer space of representation for ranking players and actions compared to Elo, a benefit we demonstrate empirically in our results.
> > > >
> > > > Can the authors elaborate on this? What are agent learning rules [1-3]?
> > > >
> > > > In the submission, the authors make an analogy to RGB pixels. However, it's not clear to me that this analogy holds up well. If I'm asked to solve a zero-sum normal-form game, I can run regret matching (among other classical algorithms) and get a good approximation in a relatively short amount of time. On the other hand, if you ask me to segment an image, there's really nothing else I can do besides deep learning.
> > > >
> > > > I agree that the elo task is compelling compared to the others. Maybe this should play a more central role in the motivation?

---

> > > > > ### Author Response · Authors · 2023-11-21
> > > > >
> > > > > Thank you for your feedback and we are glad that our revision has helped clarified the method description.
> > > > >
> > > > > Regarding your question on the motivation, let us clarify as follows:
> > > > >
> > > > > In our initial response we referred to [1-3] which is a line of earlier works in **tabular RL** learning literature where at each policy update, the agent internally constructs an NFG and solves for its equilibrium (Nash or Correlated depending on the algorithm). These NFGs correspond to Q-tables learned by the RL agents (i.e. if I were to play one action and you were to play another action, what would be our expected payoffs)? The RL policy update step then rely on the equilibria solved from such NFGs.
> > > > >
> > > > > One could note that this line of work stopped at 2003 yet there has been major advances in deep RL over the past two decades. We believe a major reason for the lack of follow-up of similar ideas, and we alluded to it in the revised introduction using NES [4] as an example, is that it is infeasible to bake in such equilibrium-solving procedure in deep RL agents' update rules --- the deep (RL) learning paradigm very much rely on the loss function being end-to-end differentiable, yet classical equilibrium solvers (or a regret matching procedure as suggested) are not end-to-end differentiable therefore one cannot use these subroutines in deep RL agents.
> > > > >
> > > > > This problem is made even worse as off-the-shelf classical solvers can fail, converge in non-deterministic time, and do not parallelise well --- in stark contrast to deep RL agents's gradient-based updates that finish in deterministic time, run on accelerator hardware with massive parallelism and must not fail. We believe our work on NfgTransformer offers a principled approach that can revive similar ideas in the age of deep (RL) learning.
> > > > >
> > > > > As you can see these tabular RL examples [1-3] are relevant but perhaps a bit more difficult to follow for an introductory paragraph. We have therefore decided to move them to the Related Works section with a brief description and used NES as an easier-to-understand example for the introduction, where [4] showed that a neural-network based equilibrium solvers can solve for millions of games in seconds using parallelism, is fully differentiable with good precision, making our main points on the benefits of an NN-based game theoretic solver.
> > > > >
> > > > > We are more than happy to revisit how we could use [1-3] as better motivating examples if you think they would strength the paper. Looking forward to hearing what you think.
> > > > >
> > > > > [1] Littman, Michael L. "Friend-or-foe Q-learning in general-sum games." ICML. Vol. 1. 2001.
> > > > >
> > > > > [2] Hu, Junling, and Michael P. Wellman. "Nash Q-learning for general-sum stochastic games." Journal of machine learning research 4.Nov (2003): 1039-1069.
> > > > >
> > > > > [3] Greenwald, Amy, Keith Hall, and Roberto Serrano. "Correlated Q-learning." ICML. Vol. 3. 2003.
> > > > >
> > > > > [4] Marris, Luke, et al. "Turbocharging solution concepts: Solving NEs, CEs and CCEs with neural equilibrium solvers." Advances in Neural Information Processing Systems 35 (2022): 5586-5600.

---

> > > > > > ### Comment · Reviewer_pter · 2023-11-22
> > > > > > **Response**
> > > > > >
> > > > > > > In our initial response we referred to [1-3] which is a line of earlier works in tabular RL learning literature where at each policy update, the agent internally constructs an NFG and solves for its equilibrium (Nash or Correlated depending on the algorithm). These NFGs correspond to Q-tables learned by the RL agents (i.e. if I were to play one action and you were to play another action, what would be our expected payoffs)? The RL policy update step then rely on the equilibria solved from such NFGs. One could note that this line of work stopped at 2003 yet there has been major advances in deep RL over the past two decades. We believe a major reason for the lack of follow-up of similar ideas, and we alluded to it in the revised introduction using NES [4] as an example, is that it is infeasible to bake in such equilibrium-solving procedure in deep RL agents' update rules --- the deep (RL) learning paradigm very much rely on the loss function being end-to-end differentiable, yet classical equilibrium solvers (or a regret matching procedure as suggested) are not end-to-end differentiable therefore one cannot use these subroutines in deep RL agents. This problem is made even worse as off-the-shelf classical solvers can fail, converge in non-deterministic time, and do not parallelise well --- in stark contrast to deep RL agents's gradient-based updates that finish in deterministic time, run on accelerator hardware with massive parallelism and must not fail. We believe our work on NfgTransformer offers a principled approach that can revive similar ideas in the age of deep (RL) learning.
> > > > > >
> > > > > > This motivation seems reasonable. Is there a reason it was not communicated in the submissions itself?
> > > > > >
> > > > > > > As you can see these tabular RL examples [1-3] are relevant but perhaps a bit more difficult to follow for an introductory paragraph. We have therefore decided to move them to the Related Works section with a brief description and used NES as an easier-to-understand example for the introduction.
> > > > > >
> > > > > > I see, I guess this is why.
> > > > > >
> > > > > > ---
> > > > > >
> > > > > > I re-read the introduction again and I think it could be improved. As is, the introduction goes (roughly):
> > > > > > 1. Representation learning is important
> > > > > > 2. Hasn't made impact in game theory
> > > > > > 3. Payoff tables are like pixels
> > > > > > 4. We do something more general than existing works
> > > > > > 5. Equivariance in normal-form games
> > > > > > 6. Equivariance desiderata
> > > > > > 7. We propose ...
> > > > > >
> > > > > > I feel like this is too abstract/meandering and doesn't directly communicate the most important thing for the introduction to communicate: the motivation for the paper. As far as revisions, to make space for this, I think (1-4) aren't that important and could be shortened or removed to make space for more direct motivations (in particular, I think that the elo experiments and the Nash Q-learning aspiration are the submission's strongest motivators).
> > > > > >
> > > > > > ---
> > > > > >
> > > > > >  Will the authors commit to releasing documented code if the submission is accepted? I think that would be helpful.

---

> > > > > > > ### Author Response · Authors · 2023-11-22
> > > > > > >
> > > > > > > > Will the authors commit to releasing documented code if the submission is accepted?
> > > > > > >
> > > > > > > Absolutely! We have every intention to release our implementation as we mentioned in our very initial submission (first page footnote) and in the global comment. We believe this would significantly help others to reproduce and extend on our work in their research which is very important for a first representation learning technique of NFGs.
> > > > > > >
> > > > > > > > I feel like this is too abstract/meandering and doesn't directly communicate the most important thing for the introduction to communicate: the motivation for the paper. As far as revisions, to make space for this, I think (1-4) aren't that important and could be shortened or removed to make space for more direct motivations.
> > > > > > >
> > > > > > > > 3. Payoff tables are like pixels
> > > > > > >
> > > > > > > Currently our point 3) motivates why representation learning could be important for NFGs. Our (perhaps subjective) opinion is that 1-3 now provide motivation and evidence on why we believe representation learning has potential in games, slightly rephrasing your summaries:
> > > > > > >
> > > > > > > 1. representation has made huge dents in other fields, vision, language etc.
> > > > > > > 2. it hasn't yet made systematic advances in game theory
> > > > > > > 3. there are early, narrow domain evidence that suggests its benefits in NFGs.
> > > > > > >
> > > > > > > Given 1-3, we look into representation learning techniques for NFGs in this work and we show indeed strong performance in a several tasks with a principled representation learning technique.
> > > > > > >
> > > > > > > We can look into revising 1-3 to make the points above clearer and incorporate Nash-Q/Correlated-Q as a motivator --- these are great suggestions and we believe it should be possible to do.
> > > > > > >
> > > > > > > We will try to upload another revision today though we respectfully disagree that 1-3 can be removed --- for a novel representation learning domain, we believe context such as 1-3 are important to readers who perhaps have not seen attempts at representation learning in NFGs. Other reviewers also shared earlier concerns on why general-purpose representation learning is needed for NFGs, and 1-3 now explain that need.
> > > > > > >
> > > > > > > **Given the upcoming deadline, we are wondering if our revision, which has significantly improved clarity and motivation compared to our initial submission, have (in full or at least in part) addressed your main concerns and if you would be willing to reconsider your initial assessment of our work?**
> > > > > > >
> > > > > > > Thank you! Revision incoming.

---

> > > > > > > > ### Author Response · Authors · 2023-11-22
> > > > > > > >
> > > > > > > > We have now revised our introduction following your suggestions: specifically, we focused on highlighting the two motivators discussed above and simplified our introduction to make room. We have also dropped the explicit statement on comparing NFGs to RGB pixels to avoid any confusion.
> > > > > > > >
> > > > > > > > 1. representation learning has made huge dents in other fields, vision, language etc.
> > > > > > > > 2. it hasn't yet made systematic advances in game theory despite many applications sharing the same data structure (i.e. NFGs)
> > > > > > > > 3. potential benefits / motivations behind representation learning of NFGs.
> > > > > > > >
> > > > > > > > We have kept this line of arguments as they provide context for designing representation learning for a novel data modality, and addresses some of other reviewers' concerns too.
> > > > > > > >
> > > > > > > > Please do let us know if this has addressed your remaining concerns and if possible, please let us know if you would be willing to revisit your initial assessment given our latest revision.
> > > > > > > >
> > > > > > > > We appreciate the reviewer's constructive feedback which has greatly improved our work compared to where it started. Thank you!

---

> > > > > > > > > ### Comment · Reviewer_pter · 2023-11-22
> > > > > > > > > **Response to Revision**
> > > > > > > > >
> > > > > > > > > I like the revision better, but I think still more could be done to clarify the motivation of the submission. As is, the reader is supposed to infer the Elo application from this sentence: "Vadori & Savani (2023) augmented this representation with learned features, enriching the player representation for much improved prediction." I feel like this could really be fleshed out into three or more sentences. I don't think I would be able to understand exactly what is being said here until after reading the Elo experiments. Also, the way the sentence is written, it gives the reader the impression that the problem has already been "solved" by Vadori & Savani (2023), rather than that it is a main motivator of the paper.

---

> > > > > > > > > > ### Author Response · Authors · 2023-11-22
> > > > > > > > > >
> > > > > > > > > > Thank you for your feedback. We have uploaded a final revision now taking into account your latest suggestions. We have removed the colour highlight but all edits are within the first three paragraphs.
> > > > > > > > > >
> > > > > > > > > > > As is, the reader is supposed to infer the Elo application from this sentence: "Vadori & Savani (2023) augmented this representation with learned features, enriching the player representation for much improved prediction."
> > > > > > > > > >
> > > > > > > > > > We have made the tasks we tackle explicit, in paragraph 3. We also explicit state that NfgTransformer brings SoTA results in all three tasks that we evaluate our method on.
> > > > > > > > > >
> > > > > > > > > > > Also, the way the sentence is written, it gives the reader the impression that the problem has already been "solved" by Vadori & Savani (2023), rather than that it is a main motivator of the paper.
> > > > > > > > > >
> > > > > > > > > > Vadori & Savani (2023) used learning in a comparatively limited way; there are dimensions of the player representation that are learned by neural networks (by minimising a specific loss they proposed), but it is not a general representation learning method. We now open the 3rd paragraph with a brief note on "recent works having used representation learning in a limited and task-specific way" when our goal is to address representation learning of NFGs explicitly and with no loss of generality.
> > > > > > > > > >
> > > > > > > > > > We hope our latest revision addresses some of your outstanding concerns. We are more than happy to make final adjustment before the end of the discussion period. Again, we appreciate your help improving our submission. Thank you!

---

> > > > > > > > ### Comment · Reviewer_pter · 2023-11-22
> > > > > > > > **Response**
> > > > > > > >
> > > > > > > > > Given the upcoming deadline, we are wondering if our revision, which has significantly improved clarity and motivation compared to our initial submission, have (in full or at least in part) addressed your main concerns and if you would be willing to reconsider your initial assessment of our work?
> > > > > > > >
> > > > > > > > Yes, the discussion period has improved my appreciation of the value of the submission. I will update my evaluation at the end of the discussion period.
> > > > > > > >
> > > > > > > > Re: 1-3, I think it's fine to include some version of the content in the submission. But I think it's a problem that a reader whose expertise is as close as mine to the subject matter of the submission could read through the introduction without actually understanding what specific problems the submission is trying to address.

---

> ### Author Response · Authors · 2023-11-21
>
> Another quick comment regarding our analogy to RGB pixels and the fact that one could solve for NE to reasonable approximation in zero-sum NFGs:
>
> In our experiments, we showed empirical results on solving for NE in 2 and **3-player** **general**-sum NFGs. The latter being more difficult but we could nonetheless approximate an NE solution to acceptable accuracy with our larger models. We can trivially extend our model to work with 4 player games too perhaps with further architectural innovations. Using classical solvers, with each generalisation of the game class (2-player 0-sum --> 2 player general-sum ---> 3 player zero-sum ...) we might need specialised tools and innovations, we think our results provide another promising approach to tackle such approximate equilibrium solving problems in a rather general fashion.
>
> NN-based solvers also have other practical benefits such as their differentiability and ease to parallelise taking advantage of accelerators to solve a large number of games in short amount of time.
>
> We think our analogy to RGB remains apt: before deep representation learning techniques revolutionised computer vision, many representation techniques have been developed too, even for tasks like segmentation or classification (e.g. Laplacian Pyramid, wavelet transform). Granted they do not achieve similar levels of performance as we can do today. Similarly, **a)** no off-the-shelf classical NE solver can solve thousands or millions of 3-player general-sum NFGs to reasonable accuracy in seconds; and **b)** we show better results on equilibrium solving than NES which relied on handcrafted feature vectors (not dissimilar to how learned representation surpassed the performance offered by laplacian pyramid/wavelet transform).
>
> Our work is only the first (principled) representation learning technique for NFGs too and our hope is that even better performance will follow with further architectural innovation as is the tradition of this conference.

---

### Official Review · Reviewer_6Yok · 2023-11-10

**Soundness:** 2 fair
**Presentation:** 1 poor
**Contribution:** 1 poor
**Rating:** 3
**Confidence:** 4

**Summary:**

The paper proposes an action-payoff encoder architecture for normal-form games (NFGs), which can be used to predict the payoffs, and maximal deviation of a player. The main contribution of this work is NfgTransformer, which is an action-payoff encoder that leverages the permutation-invariance of transformer architectures. The proposed architecture reportedly outperforms baselines in various game-theoretic toy examples.

**Strengths:**

# Strengths
- The empirical results of the proposed architecture outperforms the baselines in several toy benchmarks.

**Weaknesses:**

# Weakness
- [Presentation]
    - It is hard to parse the main goal of this work at the first glance; the main contribution of this work is a game encoder architecture that exhibits a baked-in equivariance (which is a direct consequence of transformer architecture)
    - I think it would have been much better if the authors emphasized why we need such a game encoder, and why it is important; for example, the authors state “For practical applications such as ranking in Go and Chess, it is infeasible to evaluate all pairs of players yet we may wish to make predictions about the game based on incomplete information from a subset of the matchups.” In Sec. 1 — more concrete examples would help the readers to understand the motivation of this work.
    - In Sec. 1, the authors vaguely describe the main contribution of this work as:
        - “… we consider the problem of bringing game-theoretic reasoning to deep learning systems and, conversely, using deep learning techniques to solve challenges in game theory.”
        - “… NFGs are canonical descriptions of strategic interaction between players that allows one to ask a variety of questions.”
        - But none of the descriptions clearly states that the actual contribution lies in an encoder architecture that encodes action & payoffs.
    - Hard to understand the illustration of the proposed architecture in Fig. 1 and Fig.2
        - The illustration and caption are not self-contained; readers need to resort to Sec. 4 to actually grasp the implementation
        - What is the difference between light colors and bold colors in Fig.1 ?
        - It is hard to parse the meaning of the annotations attached to the tokens in Fig. 1; for example, in “Payoffs” of Fig. 1, different lengths are identically annotated as “T”
    - Some of internal links are broken
        - Pointers to Fig. 5
        - Pointers to Proposition 3.2
        - Pointer to Sec. 5.1.
        - Pointers to Sec 4.4
        - Some links to the references
- [Technical Novelty]
    - If I understood correctly, it seems like the equivariance of NfgTransformer is a direct consequence of transformer architecture, and therefore, hard to consider it as a novel technical contribution.
- [Experiment]
    - The games presented in the experiments are rather toy-ish — given that the main contribution of this work lies in an empirical architecture, I think the efficiency of the proposed architecture should be validated on a real-world games, e.g., Go and Chess, which the authors listed as possible practical applications in Sec. 1.

**Questions:**

From my understanding, the "equivariance" property of NfgTransformer architecture is a direct consequence of transformer architecture itself -- which is OK, but hard to be considered as a significant technical contribution; it would be appreciated if the authors could clarify this point.

---

> ### Author Response · Authors · 2023-11-16
>
> We thank the reviewer for their constructive feedback and we preface our responses by stating that our initial submission has not been clear which could have added to the confusion in reviewer’s questions. We hope our revision will bring clarity to many of these questions.
>
> **Q1: permutation equivariance follows directly from the transformer architecture, please clarify your contribution.**
>
> We define permutation equivariance to be: permutation in the strategies and players of the input, results in identical permutation of the output (with *no* change to network parameters). This definition is made precise in our background section.
>
> Traditional language transformers do NOT have this property: they are causal and position aware, with appropriate masking and position embeddings. The representation after seeing the first token “dog” in “dog bites man” should be very different from the representation after seeing the last token “dog” in “man bites dog” even though we simply permuted “dog” and “man” in the input.
>
> The word "transformer" is used in NfgTransformer due to the use of self-attention and cross-attention operations, which have been popularised by the Transformer architecture. The attention operation treats key-value inputs as an unordered set; they are additionally equivariant with respect to the queries. These are the properties NfgTransformer leverages in its architecture. We now call them out explicitly in the background section on Multi-headed Attention in our revision.
>
> In short, the equivariance property of NfgTransformer is through careful design of our novel architecture proposal that is catered to the structure of NFGs, not a direct result of existing transformer architecture.
>
> **Q2: the contribution and motivation of the method should be clarified.**
>
> We agree with the reviewer that our initial introduction paragraphs were not clear and we have significantly revised our introduction paragraphs to better reflect our motivation.
>
> **Q3: presentation of the method is not clear (e.g. Fig1 and Fig2).**
>
> We recognise these issues as a major source of confusion and have revised these two figures significantly with precise notations. We have updated our description of the architecture in the text too. Please do let us know if there are other changes that would improve the presentation upon reviewing our changes.
>
> **Q4: broken links.**
> We have verified that links are working as intended in our revised PDF. Please let us know if this is not the case.
>
> **Q5: evaluation domains are toy’ish.**
> We evaluate our model on synthetic games that cover the entire equilibrium-invariant space of NFGs in Figure 3. Games sampled from this space have the property that they cover all interesting strategic interactions that could change the equilibrium solution of the NFGs at a specific game size. This is consistent with earlier work such as NES [1]. For ranking, we studied DISC games which are consistent with prior works in the ranking literature [2-3]. Finally, our interpretability results are done in GAMUT, which is the standard set of NFGs that capture relevant strategic dynamics in the economics/game theory literature.
>
> We believe our choice of test domains is principled and well-grounded in the literature targeting normal-form game theory.
>
> The normal-form payoff matrix of Go and Chess are known to be highly transitive [4] and we would expect them to tend to have dominant actions (players) and therefore make for less comprehensive evaluation.
>
> [1] Marris, Luke, et al. "Turbocharging solution concepts: Solving NEs, CEs and CCEs with neural equilibrium solvers." Advances in Neural Information Processing Systems 35 (2022): 5586-5600.
>
> [2] Bertrand, Quentin, Wojciech Marian Czarnecki, and Gauthier Gidel. "On the Limitations of the Elo, Real-World Games are Transitive, not Additive." International Conference on Artificial Intelligence and Statistics. PMLR, 2023.
>
> [3] Vadori, Nelson, and Rahul Savani. "Ordinal Potential-based Player Rating." arXiv preprint arXiv:2306.05366 (2023).
>
> [4] Czarnecki, Wojciech M., et al. "Real world games look like spinning tops." Advances in Neural Information Processing Systems 33 (2020): 17443-17454.

---

> > ### Author Response · Authors · 2023-11-21
> > **Discussion period ending soon**
> >
> > Dear reviewer 6Yok,
> >
> > As we are approaching the end of the discussion period we are wondering if you have had a chance to review our significant revision and our responses above? If so, please do let us know if our revision has addressed your questions and concerns.
> >
> > Thank you!

---

### Author Response · Authors · 2023-11-16

We thank all reviewers for their constructive feedback which has helped improve our work significantly. We have since made the following major points of revision to the paper, with important changes highlighted in blue in the updated PDF:

* We have reframed the introduction paragraph to better clarify the motivation behind our work.
* We have reworked Figure 1 and Figure 2, which now describe different variables involved in each step of the computation in the encoder, decoder and loss functions in precise terms.

We recognise that our initial presentation was abstract and did not convey our ideas and contributions clearly. We hope the changes we made would help clarify our proposal as well as the motivation behind our work.

We also plan to release open-source implementations of our proposed architecture upon publication which could further clarify our method.

We have replied to each reviewer in specifics in threads below.

---

### Meta-Review · Area_Chair_cBDg · 2023-12-06

**Metareview:**

The signal in reviewer 6Yok's review is rather weak, and given that they have remained unengaged, I will appropriately discount it.

Overall, I believe the paper is a good contribution, and I will go against the crude average score here to recommend presentation as a poster or spotlight.

In addition to the reviewer's comment, I would appreciate it if you could make Figure 4 more readable (the font is waaay too small), and invest some time into polishing the presentation along the directions recommended by the reviewers (especially #pter).

Also, please do not forget you have promised to open source the code.

**Justification For Why Not Higher Score:**

I do not this the contribution is the top 5% of paper, however this is a good paper and could be presented as spotlight.

**Justification For Why Not Lower Score:**

I think that the paper deserves, at a minimum, acceptance.

---

### Decision · Program_Chairs · 2024-01-16

Accept (poster)